# BAYESIAN-LoRA: GAUSSIAN PROCESS MODELING FOR LARGE LANGUAGE MODELS

## ABSTRACT

Pre-trained LLMs are often reasonably calibrated on pre-training like distributions, but fine-tuning them for specific domains often causes a substantial deterioration in calibration, especially on small datasets, leading to overconfident predictions. Although existing Bayesian approaches alleviate this degradation, their computational cost is prohibitive at LLM scale. In this work, we introduce **Bayesian-LoRA**, which applies a Sparse Gaussian Process (SGP) to the Low-Rank Adaptation (LoRA) fine-tuning approach and integrates a normalizing flow to stabilize the training process, thereby substantially improving calibration in fine-tuned LLMs. We conduct extensive experiments on the LLaMA 2-7B model across a set of commonsense reasoning benchmarks. With only approximately 0.42M additional parameters over LoRA, Bayesian-LoRA reduces the calibration error (ECE) and Negative Log-Likelihood (NLL) without sacrificing accuracy, for both in-distribution and out-of-distribution (OOD) evaluations, while retaining the LoRA's parameter efficiency and incurring only modest extra training/memory overhead.

## 1 INTRODUCTION

Fine-tuning of large language models (LLMs) (Ding et al., 2023; Xu et al., 2021; Malladi et al., 2023; Hu et al., 2022) has become increasingly prevalent across a wide range of domains (Hu et al., 2024; Wang et al., 2024b; Shorinwa et al., 2025). It adapts a powerful general-purpose LLM to specific downstream tasks with substantially reduced computational cost, memory footprint, and data requirements (Yin et al., 2024; Lin et al., 2024; Liu et al., 2025; Ye et al., 2024).

However, although pre-trained models are reasonably well calibrated out of the box (Zhou et al., 2024; Wang, 2023), calibration often deteriorates after domain-specific fine-tuning and as a result, models tend to become systematically overconfident (Liu et al., 2024; Mai et al., 2024; Zhou et al., 2023). Such degradation is unacceptable when LLMs are applied in safety-critical domains, such as autonomous driving (Tu et al., 2025; Wu et al., 2021), medical diagnosis (Savage et al., 2025), and many others (Liu et al., 2025; Kim et al., 2025; Chuang et al., 2025). Therefore, there is an urgent need for calibration-aware fine-tuning methods that can maintain or improve probability calibration during LLM training.

Probabilistic approaches, such as Bayesian neural networks with Laplace posterior approximations (Yang et al., 2024) or stochastic formulations (Lin et al., 2025a), provide a principled way to estimate uncertainty that can generate better-calibrated probabilities. But applying these probabilistic principles to modern LLMs is computationally prohibitive (Daxberger et al., 2021; Wang et al., 2024a), as posterior inference is costly and calibration often underperforms under distribution shift. Sparse Gaussian Processes (SGPs) provide a scalable approximation to Gaussian processes by introducing a small set of inducing variables, and thereby keeping uncertainty quantification and preserving inference tractability for large-scale models (Titsias, 2009; Burt et al., 2020).

Existing research often either applies probabilistic methods during the pre-training stage, which is infeasible at LLM scale (Xue et al., 2021; Sankararaman et al., 2022; Zhao et al., 2025), or adopts post-hoc calibration after fine-tuning (Yang et al., 2024), which cannot fundamentally prevent the degradation of calibration during training. Specifically, LA/LLLA (Yang et al., 2024) performs Laplace approximation (Gaussian second-order approximation) on the parameter posterior near the MAP point that refers to the deterministic LoRA, while BLoB (Wang et al., 2024a) places explicit

priors on the LoRA weights and infers their posterior with the variational inference in the parameter space, both of which mainly attribute the uncertainty to parameter perturbations. However, both approaches rely on parameter-space posterior approximation, which brings with it the key disadvantage of tying uncertainty to parameterization and local Hessians that could be solved using function-space approximations. However, to the best of our knowledge, this has not yet been explored in the literature.

In this work, we propose Bayesian-LoRA, a calibration-aware fine-tuning framework that integrates a sparse Gaussian process into the LoRA (Hu et al., 2022) adaptation scheme, which allows efficient uncertainty estimation and preserves the parameter efficiency of LoRA. We perform an extensive empirical investigation on the effectiveness of our method by fine-tuning LLaMA 2-7B on six widely used commonsense reasoning benchmarks, including WinoGrande-S (Sakaguchi et al., 2020), ARC-C (Clark et al., 2018), ARC-E (Clark et al., 2018), WinoGrande-M (Sakaguchi et al., 2020), OBQA (Mihaylov et al., 2018), and BoolQ (Clark et al., 2019). Beyond multiple-choice settings, our approach also performs strongly on generative language modeling on the WikiText-2 (Merity et al., 2016) dataset. These experiments demonstrate that our Bayesian-LoRA achieves almost SOTA performance across diverse benchmarks. For the details, on ARC-E our method reduces ECE by up to 37% over LoRA; on BoolQ it improves NLL by around 15% and halves the calibration error; and on WikiText-2 it further lowers both NLL and ECE, while adding only $\approx$ **0.42M** parameters over LoRA, and the training time/memory are close to LoRA ($\approx 1.2\times$).

## 1.1 RELATED WORK

Model calibration is of crucial importance for trustworthy machine learning. Although modern neural networks can achieve high predictive accuracy, they have often been found to be poorly calibrated and tend to produce overconfident predictions (Guo et al., 2017). In large language models (LLMs), these issues become more pronounced, particularly when fine-tuning for domain-specific tasks by updating the pre-trained model's weights based on a small dataset, since this often amplifies calibration bias problems (Liu et al., 2024; Mai et al., 2024). In calibrated models, predicted probabilities match the actual frequency of correctness. Poor calibration leads to overconfident or misleading confidence estimates, reducing reliability in downstream tasks.

Post-hoc methods improve probability calibration by scaling model predictions after training (Kull et al., 2019; Guo et al., 2017). However, these methods cannot fundamentally prevent the degradation of uncertainty estimation in model weights, as they just rescale the output distribution after training. And importantly, when the input distribution shifts or the model encounters out-of-distribution data, the effectiveness of post-hoc calibration often declines dramatically (Desai & Durrett, 2020).

Bayesian methods provide a principled framework to quantify model uncertainty, including Bayesian Neural Networks (Izmailov et al., 2021; Lin et al., 2025a; Kweon et al., 2025), Laplace approximations (Yang et al., 2024), Gaussian processes (Ranković & Schwaller, 2025), and many others (Abbasi Yadkori et al., 2024; Rajamohan et al., 2025). These methods have been shown to be useful tools to improve calibration and robustness in small-scale models. However, when it comes to Large Language Models (LLMs), they incur prohibitive computational cost. For example, Full-model Bayesian (each weight has a distribution) needs to make an approximate inference over entire parameter spaces (Graves, 2011; Blundell et al., 2015). For instance, the work of Wang et al. on BLoB (Wang et al., 2024a) applied this concept, but also inherits its drawbacks in LLMs. Similarly, Laplace methods (LA/LLLA) (Yang et al., 2024) always depend on estimating and storing (block) Hessians, whose cost grows rapidly with model size (Daxberger et al., 2021; Ritter et al., 2018). Gaussian processes (with sparsity) need cubic-level time and quadratic memory consumption in the number of data points (Seeger, 2004; Quinonero-Candela & Rasmussen, 2005), and ensemble techniques multiply inference-time computation (Lakshminarayanan et al., 2017). Instead, we model the uncertainty constraints within the low-rank subspace of LoRA, combining sparse GP with normalizing flows to induce $\Delta\mathbf{W}$ with low overhead.

Parameter-Efficient Fine-Tuning (PEFT) uses a small set of additional parameters and keeps the base LLM structure to reduce fine-tuning computational cost and memory footprint. It has been proven successful and is widely applied in industry. Some prominent methods, e.g., adapter-based tuning (Adapters/AdapterFusion) (Houlsby et al., 2019; Pfeiffer et al., 2021), prompt/prefix tuning (Lester et al., 2021; Li & Liang, 2021), and low-rank adaptation (LoRA and its variants) (Hu et al., 2022; Zhang et al., 2023; Dettmers et al., 2023) are adapted in many areas. However, PEFT techniques

employ deterministic updates (e.g., LoRA's deterministic low-rank increments) and do not explicitly model uncertainty or optimize calibration objectives within the adapter subspace.

In this context, we propose a sparse-Gaussian-process-based Bayesian modeling framework, which we combine with a LoRA mechanism in low-rank subspace to adapt probabilistic principles to LLMs and also keep the parameter and memory efficiency of PEFT.

## 2 PRELIMINARIES

This section provides the necessary background. Firstly, we review the fundamental concepts of LoRA, parameterize the uncertainty of $\Delta\mathbf{W}$, and briefly describe the required sparse GP and normalizing flow foundations.

### 2.1 LoRA: LOW-RANK ADAPTATION

Low-rank adaptation (LoRA) (Hu et al., 2022) adopts a low-rank parameterization, where $\Delta\mathbf{W}$ is factorized as $\frac{\alpha}{r}\mathbf{B}\mathbf{A}$ with rank $r \ll \min(d_{\text{in}}, d_{\text{out}})$ and keeps the base model's weight, $\mathbf{W} \in \mathbb{R}^{d_{\text{out}} \times d_{\text{in}}}$, frozen during the training, where only $\mathbf{A}$ and $\mathbf{B}$ are trained, and $\mathbf{A}$ is usually initialized to zero and uses a scaling $\alpha/r$ to control the update magnitude. This is usually applied to Transformer projection layers (e.g., $W_q, W_k, W_v, W_o$) and can be merged into $\mathbf{W}$ with negligible added latency:

$$\Delta\mathbf{W} = \frac{\alpha}{r}\mathbf{B}\mathbf{A}, \qquad \mathbf{B} \in \mathbb{R}^{d_{\text{out}} \times r}, \ \mathbf{A} \in \mathbb{R}^{r \times d_{\text{in}}} \tag{1}$$

$$\mathbf{y} = (\mathbf{W} + \Delta\mathbf{W})\mathbf{x} = \mathbf{W}\mathbf{x} + \frac{\alpha}{r}\mathbf{B}(\mathbf{A}\mathbf{x}) \tag{2}$$

The number of trainable parameters in LoRA is $\text{Params}_{\text{LoRA}} = r(d_{\text{in}} + d_{\text{out}})$, compared to $d_{\text{in}}d_{\text{out}}$ for full fine-tuning, resulting in savings at a ratio of $\frac{r(d_{\text{in}}+d_{\text{out}})}{d_{\text{in}}d_{\text{out}}}$.

### 2.2 VARIATIONAL SPARSE INDUCING WEIGHT MODEL

Consider the weight matrix of a neural network layer to be $W \in \mathbb{R}^{d_{\text{out}} \times d_{\text{in}}}$. Instead of defining a variational distribution directly on $W$, we introduce a low-dimensional inducing matrix: $U \in \mathbb{R}^{r \times c}, r \ll d_{\text{out}}, c \ll d_{\text{in}}$ to serve as a set of variational parameters controlling the distribution over $W$ (Yang et al., 2024; Snelson & Ghahramani, 2005). We set a Gaussian prior for $U$:

$$p(U) = \mathcal{N}(\text{vec}(U) \mid \mathbf{0}, \mathbf{K}_U), \qquad \mathbf{K}_U = \mathbf{K}_c \otimes \mathbf{K}_r, \tag{3}$$

where $\mathbf{K}_r \in \mathbb{R}^{r \times r}$ and $\mathbf{K}_c \in \mathbb{R}^{c \times c}$ are the row and column side covariances, and $\otimes$ denotes the Kronecker product. We then assume a Gaussian variational posterior on $U$, $q(U) = \mathcal{N}(\text{vec}(U) \mid \mathbf{m}, \mathbf{S})$ where $\mathbf{m}$ and $\mathbf{S}$ denote the mean vector and covariance matrix. Whitening can be applied such that the prior on $U$ becomes standard normal.

Given inducing variables $U$, the Gaussian conditional distribution (Lin et al., 2025a) of the weight matrix is:

$$p(W \mid U) = \mathcal{N}(W \mid M_W(U), \ \lambda^2 \Sigma_W) \tag{4}$$

where $\lambda > 0$ is a scaling parameter. And $\Sigma_W$ represents the variance of $W$. Define the row and column inducing covariances:

$$K_r = Z_r Z_r^\top + D_r^2, \qquad K_c = Z_c Z_c^\top + D_c^2 \tag{5}$$

with corresponding projection operators $\mathbf{T_r} = Z_r^\top K_r^{-1}$ and $\mathbf{T_c} = K_c^{-1} Z_c$ . The conditional mean (**Weight**) is then expressed as (for details see Appendix A):

$$M_W(U) = T_r U T_c \tag{6}$$

The marginal distribution over weights is obtained by integrating out $U$: $q(W) = \int p(W \mid U) q(U) \, dU$. Thus, optimizing only the variational distribution of the low-dimensional inducing variables suffices to approximate the high-dimensional posterior over $W$ (Lin et al., 2025b). The variational evidence lower bound is:

$$\mathcal{L} = \mathbb{E}_{q(W)}\big[\log p(\mathcal{D} \mid W)\big] - \text{KL}\big(q(U) \,\|\, p(U)\big) - \mathbb{E}_{q(U)}\Big[\text{KL}\big(q(W \mid U) \,\|\, p(W \mid U)\big)\Big] \tag{7}$$

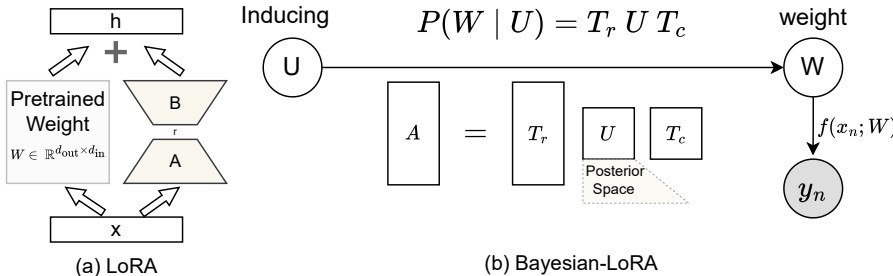

Figure 1: Graphical overview of our proposed framework **Bayesian-LoRA**. Inducing variables $U$ generate the effective weights A and B through a conditional Gaussian distribution $p(W_A \mid U)$ and $p(W_B \mid U)$. We have also provided an alternative implementation in Appendix B, Figure 4

where the first term is the expected log-likelihood, the second is the KL divergence between the variational posterior and prior of the inducing variables, and the third is the conditional KL divergence. Both $q(W \mid U)$ and $p(W \mid U)$ share the same mean and differ only by a covariance scaling, and the conditional KL keeps a closed-form expression.

## 3 METHODOLOGY: BAYESIAN-LORA

We replace the deterministic low-rank update of LoRA with a probabilistic low-rank induced weight. Set the weight matrix of a layer to be $W \in \mathbb{R}^{d_{\text{out}} \times d_{\text{in}}}$. Instead of directly performing variational optimization on $W$, we introduce a low-dimensional induced matrix $U \in \mathbb{R}^{r \times c}$ (where $r \ll d_{\text{out}}$ and $c \ll d_{\text{in}}$), and "diffuse" low-rank information in the high-dimensional weight space via a conditional Gaussian $p(W \mid U)$. Compared to LoRA's deterministic update $\Delta W = \frac{\alpha}{r} BA$, Bayesian-LoRA randomizes this update and quantifies uncertainty: $\Delta W$ is induced by the posterior uncertainty of $U$.

### 3.1 LORA TO BAYESIAN-LORA

LoRA uses a rank-$r$ update

$$\Delta W_{\text{LoRA}} = \frac{\alpha}{r} BA, \qquad B \in \mathbb{R}^{d_{\text{out}} \times r}, \ A \in \mathbb{R}^{r \times d_{\text{in}}} \tag{8}$$

In our model, we follow Eq. 3, Eq. 4, Eq. 5, and Eq. 6 to model the uncertainty of LoRA weight in low-rank subspace.

However, a purely Gaussian prior can be overly restrictive in the low-rank space and underperforms in capturing the whole weight-space uncertainty. To account for posterior complexity and keep parameters light, inspired by (Lin et al., 2025b), we enrich the family by placing a normalizing flow on top of a diagonal-Gaussian base, which takes flexible, non-Gaussian uncertainty over $U$ and better coverage of the weight-space variability.
We place a Gaussian (matrix-normal) prior on $U$: $p(U) = \mathcal{N}\big(\text{vec}(U) \mid \mathbf{0}, \mathbf{K}_c \otimes \mathbf{K}_r\big)$. We then put a *normalizing flow* on top of a diagonal-Gaussian base:

$$q_0(U_0) = \mathcal{N}\big(\text{vec}(U_0) \mid \mathbf{m}, \text{diag}(\boldsymbol{\sigma}^2)\big), \qquad U = T_\phi(U_0) \tag{9}$$

Here, $T_\phi$ is an invertible, differentiable map; in practice, we use lightweight row-wise Masked Autoregressive Flow (MAF). By change of variables,

$$\log q_\phi(U) = \log q_0\big(T_\phi^{-1}(U)\big) - \log\Big| \det J_{T_\phi}\big(T_\phi^{-1}(U)\big)\Big| \tag{10}$$

When $T_\phi$ is the identity, $q_\phi(U)$ reduces to the diagonal-Gaussian baseline; increasing flow capacity improves posterior fit and downstream calibration.

Substituting the $q(\mathbf{u})$ (Eq.10) into Eq. 7, the standard SVGP Evidence Lower Bound (ELBO), the ELBO becomes:

$$\mathcal{L}_{\text{ELBO}} = \mathbb{E}_{U_0 \sim q_0, \, \epsilon}\big[ \log p(\mathcal{D} \mid W)\big]$$

$$- \mathbb{E}_{U_0 \sim q_0}\Big[ \log q_0(U_0) - \log\big| \det J_{T_\phi}(U_0)\big| - \log p\big(T_\phi(U_0)\big)\Big] - \frac{D}{2}\Big( \lambda^2 - 1 - 2\log \lambda \Big) \tag{11}$$

where $D = \sum_\ell d_W^{(\ell)}$ represents the sum of the weight dimensions of all replaced layers. To obtain Eq. 11, we replace the variational posterior $q(U)$ with the flow-based distribution $q_\phi(U)$ defined in Eq. 9 and Eq. 10. Meanwhile, the conditional KL term $\mathbb{E}_{q(U)}[\mathrm{KL}(q(W|U)|p(W|U))]$ admits a closed form, independent of $U$, which outputs $\frac{D}{2}(\lambda^2 - 1 - 2\log \lambda)$. This introduces the additional Jacobian determinant term $\log \det J_{T_\phi}(U_0)$.

By Proposition 3.1 (KL invariance under $T_\phi$), the ELBO written in $U$ equals that in $\Delta W$, so we may optimize in $U$-space while evaluating in weight space (proof in Appendix A.5, Proposition A.1).

**Proposition 3.1** (KL invariance under $T_\phi$). *Let $T_\phi$ be invertible and set $\Delta W = T_\phi(U)$. With pushforwards $q_\phi := T_{\phi\#}q_\psi$ and $p_\phi := T_{\phi\#}p$, we have*

$$\mathrm{KL}\big(q_\phi \,\|\, p_\phi\big) = \mathrm{KL}\big(q_\psi \,\|\, p\big). \tag{12}$$

*Hence, the ELBO written in $U$-space equals the ELBO written in $\Delta W$. Proof. See Appendix A.1.*

---

**Algorithm 1** Bayesian-LoRA (replace $A, B$): One Training Step

---

**Require:** Batch $\mathcal{B}$, target layers $\mathcal{L}$, rank $r$, scale $\alpha$, MC $S$, noise $\lambda$, flow $T_\phi$, base $q_0$, per-layer $(K_r^{A/B}, K_c^{A/B}, T_r^{A/B}, T_c^{A/B})$
1: Precompute per-layer caches $C_\ell^{A/B}$ using $(K_r^{A/B}, K_c^{A/B}, T_r^{A/B}, T_c^{A/B})$     ▷ Eqs. 5-9
2: **for** $\ell \in \mathcal{L}$ **do**             ▷ broadcast over $s = 1{:}S$
3:   Sample $U_0^{(1:S)} \sim q_0$; set $U^{(1:S)} \leftarrow T_\phi(U_0^{(1:S)})$
4:   $\bar{A}_\ell^{(s)} = T_{r,\ell}^A U^{(s)} T_{c,\ell}^A, \quad \bar{B}_\ell^{(s)} = T_{r,\ell}^B U^{(s)} T_{c,\ell}^B$      ▷ Eq. 6
5:   $A_\ell^{(s)} = \bar{A}_\ell^{(s)} + \lambda \Sigma_{A,\ell}^{1/2}\varepsilon_A^{(s)}, \quad B_\ell^{(s)} = \bar{B}_\ell^{(s)} + \lambda \Sigma_{B,\ell}^{1/2}\varepsilon_B^{(s)}$
6:   $\Delta W_\ell^{(s)} = \frac{\alpha}{r} B_\ell^{(s)} A_\ell^{(s)}, \quad W_{\mathrm{eff},\ell}^{(s)} = W_{\mathrm{pre},\ell} + \Delta W_\ell^{(s)}$     ▷ Eq. 8
7: **end for**
8: Compute $\mathcal{L}_{\mathrm{ELBO}}$ on $\mathcal{B}$ using $\{W_{\mathrm{eff},\ell}^{(s)}\}$        ▷ Eqs. 10-11

---

As shown in Figure 1, the LoRA weights, $W_{\mathrm{LoRA}}$, are now generated by a sparse Gaussian process following Eq. 6. In this view, $T_r$ is the left covariance of $U$, and $T_c$ the right covariance. We present pseudocode in Algorithm 1.

## 4 EXPERIMENTS

We chose **LLaMA 2-7B** as the base backbone with an adapter of LoRA on the attention layers in Queries (Q), Key (K), and LM Head weight matrices. In particular, we used the PEFT library (Mangrulkar et al., 2022) to initialize the LoRA and replaced the corresponding linear layers with a Bayesian implementation. The details and notation follow Sec. 3. We evaluate on six commonsense reasoning benchmarks (datasets and preprocessing in Appendix D). All comparison methods use the same data splits, wall-clock budget, and decoding settings, and each configuration is run with three random seeds and we report mean $\pm$ std. We report the metrics Accuracy (ACC ↑), Expected Calibration Error (ECE ↓; 15 bins), and Negative Log-Likelihood (NLL ↓). This section empirically tests three claims of *Bayesian-LoRA*: 1) It improves accuracy under a compute budget comparable to standard LoRA; 2) It markedly improves calibration and likelihood; 3) It offers a superior cost-effectiveness trade-off between computational cost and performance.

Accordingly, we present a systematic evaluation on six commonsense reasoning benchmarks with respect to **performance** (Table 1), **efficiency analysis** (Table 4), detailed **ablations** (Table 5), **robustness** (Table 6), and **Bayesian Optimization for hyperparameters** (Tables 7 and 14).

### 4.1 EVALUATION UNDER IN-DISTRIBUTION SETTING

**Bayesian-LoRA** provides a structure that is naturally aligned with LoRA's low-rank concepts. In Figure 1, we replace LoRA matrices A and B with a Sparse Gaussian Process with a Kronecker structure. The details of hyperparameters are provided in Appendix F. Importantly, post-hoc work like LA (Yang et al., 2024) needs to compute the gradient of the output with respect to the parameters. This is affordable for simple multiple-choice tasks that involve only a few tokens, but

it becomes impractical for next-token prediction as thousands of backpropagations are computationally prohibitive. In contrast, **Bayesian-LoRA** does not require gradient-based post-hoc Hessian computation; instead, it estimates uncertainty end-to-end during training. In Table 1, we compared

Table 1: Results on six commonsense reasoning benchmarks. We report Accuracy (ACC ↑), Expected Calibration Error (ECE ↓; 15 bins), and Negative Log-Likelihood (NLL ↓) at a unified comparison point, the early stop checkpoint of each comparison method. Values are mean±std over three seeds. Results are with inducing points of $r_{ind} = c_{ind} = 9$ that same with MAP(LoRA), for a fair comparison.

| Metrics | Methods | WG-S | ARC-C | ARC-E | WG-M | OBQA | BoolQ |
|---|---|---|---|---|---|---|---|
| ACC ↑ | MAP (Hu et al., 2022) | $68.00 \pm 0.21$ | $64.90 \pm 1.10$ | $85.20 \pm 0.60$ | $73.70 \pm 0.90$ | $77.70 \pm 0.80$ | $85.80 \pm 0.40$ |
| | Dropout (Gal & Ghahramani, 2016) | $66.70 \pm 0.30$ | $64.90 \pm 1.90$ | $85.10 \pm 0.50$ | $73.50 \pm 0.90$ | $77.70 \pm 0.20$ | $85.90 \pm 0.40$ |
| | Ckpt Ens (Huang et al., 2017) | $66.70 \pm 0.30$ | $64.90 \pm 1.10$ | $85.20 \pm 0.60$ | $73.80 \pm 1.00$ | $78.20 \pm 0.20$ | $85.40 \pm 0.30$ |
| | Temp (Guo et al., 2017) | $67.00 \pm 0.60$ | $64.90 \pm 1.10$ | $85.20 \pm 0.60$ | $73.70 \pm 0.90$ | $77.70 \pm 0.80$ | $85.80 \pm 0.40$ |
| | BBB (Blundell et al., 2015) | $56.54 \pm 7.87$ | $68.13 \pm 1.27$ | $77.0 \pm 0.2$ | $73.63 \pm 2.44$ | $77.02 \pm 0.2$ | $83.17 \pm 0.53$ |
| | LLLA (post-hoc) (Yang et al., 2024) | $66.90 \pm 0.50$ | $66.10 \pm 0.60$ | $84.80 \pm 0.50$ | $73.70 \pm 0.90$ | $77.60 \pm 0.70$ | $85.80 \pm 0.40$ |
| | LA (post-hoc) (Yang et al., 2024) | $66.90 \pm 0.60$ | $66.90 \pm 1.10$ | $85.40 \pm 0.40$ | $73.70 \pm 1.00$ | $78.10 \pm 0.70$ | $85.80 \pm 0.40$ |
| | BLoB (N=10) (Wang et al., 2024a) | $69.07 \pm 0.34$ | $68.81 \pm 1.09$ | $85.56 \pm 0.35$ | $73.69 \pm 0.17$ | $81.52 \pm 0.74$ | $\mathbf{86.99 \pm 0.24}$ |
| | Bayesian-LoRA (S = 4) (End-to-End) | $\mathbf{70.90 \pm 0.1}$ | $\mathbf{68.90 \pm 0.2}$ | $\mathbf{85.91 \pm 0.3}$ | $\mathbf{74.30 \pm 0.2}$ | $81.60 \pm 0.1$ | $86.10 \pm 0.2$ |
| ECE ↓ | MAP (Hu et al., 2022) | $30.80 \pm 1.80$ | $26.10 \pm 1.40$ | $8.90 \pm 0.30$ | $24.90 \pm 1.30$ | $9.80 \pm 1.00$ | $7.40 \pm 0.10$ |
| | Dropout (Gal & Ghahramani, 2016) | $29.50 \pm 1.60$ | $25.60 \pm 0.70$ | $8.80 \pm 0.60$ | $23.50 \pm 1.20$ | $8.80 \pm 0.80$ | $7.50 \pm 0.10$ |
| | Ckpt Ens (Huang et al., 2017) | $25.20 \pm 1.60$ | $26.10 \pm 1.40$ | $8.90 \pm 0.30$ | $22.80 \pm 1.40$ | $4.70 \pm 0.50$ | $3.20 \pm 0.50$ |
| | Temp (Guo et al., 2017) | $12.80 \pm 0.90$ | $\mathbf{4.60 \pm 1.00}$ | $4.70 \pm 0.80$ | $6.30 \pm 1.60$ | $7.20 \pm 2.60$ | $2.50 \pm 0.30$ |
| | BBB (Blundell et al., 2015) | $21.81 \pm 12.95$ | $26.23 \pm 1.47$ | $12.28 \pm 0.58$ | $15.76 \pm 4.71$ | $11.38 \pm 1.07$ | $3.74 \pm 0.10$ |
| | LLLA (post-hoc) (Yang et al., 2024) | $11.60 \pm 1.30$ | $5.60 \pm 2.10$ | $4.20 \pm 0.30$ | $3.80 \pm 1.40$ | $5.40 \pm 0.40$ | $1.70 \pm 0.50$ |
| | LA (post-hoc) (Yang et al., 2024) | $7.80 \pm 1.90$ | $7.50 \pm 1.20$ | $\mathbf{3.40 \pm 0.80}$ | $4.80 \pm 1.60$ | $\mathbf{3.50 \pm 0.40}$ | $1.90 \pm 0.30$ |
| | BLoB (N=10) (Wang et al., 2024a) | $9.35 \pm 1.37$ | $9.59 \pm 1.88$ | $3.64 \pm 0.53$ | $3.01 \pm 0.12$ | $3.77 \pm 1.47$ | $\mathbf{1.41 \pm 0.19}$ |
| | Bayesian-LoRA (S = 4) (End-to-End) | $\mathbf{4.90 \pm 0.20}$ | $9.20 \pm 0.70$ | $5.30 \pm 0.10$ | $\mathbf{3.00 \pm 0.10}$ | $5.70 \pm 0.20$ | $2.10 \pm 0.60$ |
| NLL ↓ | MAP (Hu et al., 2022) | $2.75 \pm 0.57$ | $1.64 \pm 0.19$ | $0.54 \pm 0.03$ | $2.43 \pm 0.50$ | $0.71 \pm 0.03$ | $0.43 \pm 0.01$ |
| | Dropout (Gal & Ghahramani, 2016) | $2.54 \pm 0.49$ | $1.55 \pm 0.16$ | $0.52 \pm 0.04$ | $2.12 \pm 0.35$ | $0.71 \pm 0.04$ | $0.43 \pm 0.01$ |
| | Ckpt Ens (Huang et al., 2017) | $1.31 \pm 0.04$ | $1.64 \pm 0.18$ | $0.54 \pm 0.03$ | $1.89 \pm 0.24$ | $0.65 \pm 0.02$ | $0.35 \pm 0.01$ |
| | Temp (Guo et al., 2017) | $0.68 \pm 0.01$ | $0.90 \pm 0.01$ | $0.43 \pm 0.02$ | $0.58 \pm 0.01$ | $0.67 \pm 0.02$ | $0.35 \pm 0.00$ |
| | BBB (Blundell et al., 2015) | $1.40 \pm 0.55$ | $2.23 \pm 0.04$ | $0.91 \pm 0.06$ | $0.84 \pm 0.15$ | $0.66 \pm 0.05$ | $0.31 \pm 0.00$ |
| | LLLA (post-hoc) (Yang et al., 2024) | $0.68 \pm 0.01$ | $0.94 \pm 0.02$ | $0.44 \pm 0.01$ | $0.56 \pm 0.01$ | $0.66 \pm 0.02$ | $0.35 \pm 0.00$ |
| | LA (post-hoc) (Yang et al., 2024) | $0.66 \pm 0.02$ | $0.86 \pm 0.02$ | $0.41 \pm 0.02$ | $0.55 \pm 0.01$ | $0.62 \pm 0.01$ | $0.34 \pm 0.00$ |
| | BLoB (N=10) (Wang et al., 2024a) | $\mathbf{0.63 \pm 0.01}$ | $0.78 \pm 0.02$ | $0.40 \pm 0.01$ | $\mathbf{0.54 \pm 0.00}$ | $0.50 \pm 0.01$ | $0.31 \pm 0.00$ |
| | Bayesian-LoRA (S = 4) (End-to-End) | $0.79 \pm 0.01$ | $\mathbf{0.77 \pm 0.05}$ | $\mathbf{0.38 \pm 0.09}$ | $0.62 \pm 0.11$ | $\mathbf{0.49 \pm 0.02}$ | $\mathbf{0.29 \pm 0.01}$ |

the performance of multiple methods on six commonsense reasoning benchmarks. Overall, traditional LoRA (MAP) (Hu et al., 2022) performs well in accuracy but has significant shortcomings in calibration metrics (ECE and NLL). Post-hoc methods (Guo et al., 2017; Yang et al., 2024), such as temperature scaling and Laplace approximation, can improve the calibration of the model to some extent but do not bring significant accuracy improvements. Simple Bayesian methods, such as BLoB (Wang et al., 2024a) and BBB (Blundell et al., 2015), achieve high accuracy and low negative log likelihood on some datasets that demonstrate relatively good performance. In contrast, **Bayesian-LoRA** achieves optimal or near-optimal accuracy on most datasets and demonstrates competitive calibration performance in multiple scenarios.

Beyond multiple-choice settings, our approach also performs strongly on generative language modeling on WikiText-2 (Merity et al., 2016) dataset. As shown in Table 3, we also achieve the best performance in generative language modeling on WikiText-2, where Bayesian-LoRA ($S$=2) outperforms all baselines in both NLL and calibration metrics.

## 4.2 BEYOND SMALL MODELS: VALIDATION ON LARGER ARCHITECTURES

To ensure fair comparison with prior baselines, our earlier experiments were conducted on relatively small model configurations that matched the parameter scales and hyperparameter settings reported in previous work, which does not reveal the effectiveness of **Bayesian-LoRA**. In this section, we conduct more comprehensive experiments on larger architectures, **Qwen2.5-14B-Instruct** and **Qwen3-30B-A3B-Instruct-2507** in reasoning tasks. Table 2 shows that **Bayesian-LoRA** pro-

Table 2: Performance comparison on the **MATH** dataset using large-scale models. We report **CoT Negative Log-Likelihood (NLL)**, **CoT Expected Calibration Error (ECE)**, and **final answer accuracy**. Baseline results come from standard fine-tuning, and our Bayesian-LoRA approach. The hyperparameters for training configuration are reported in Table 15.

| Model (Zero-shot) | Method | CoT-NLL ↓ | CoT-ECE ↓ | Answer Acc. ↑ |
|---|---|---|---|---|
| Qwen2.5-14B-Instruct | Baseline FT | 2.165 | 12.2 | 49.8 |
| | **Bayesian-LoRA** | **0.513** | **5.81** | **51.1** |
| Qwen3-30B-A3B-Instruct-2507 | Baseline FT | 1.096 | 8.96 | 61.8 |
| | **Bayesian-LoRA** | 0.721 | 6.32 | 61.9 |

Table 3: Language modeling on **WikiText-2** (validation/test). We report perplexity NLL ↓, Brier ↓, and ECE ↓ (15 bins). Results are shown for both *all tokens* and the *top 5% most uncertain tokens from MAP* (selected by predictive entropy).

| | All tokens | | | | | | Top 5% entropy tokens | | | | | |
|---|---|---|---|---|---|---|---|---|---|---|---|---|
| | Validation | | | Test | | | Validation | | | Test | | |
| Method | NLL ↓ | Brier ↓ | ECE ↓ | NLL ↓ | Brier ↓ | ECE ↓ | NLL ↓ | Brier ↓ | ECE ↓ | NLL ↓ | Brier ↓ | ECE ↓ |
| LoRA (MAP) | 1.76 | 0.52 | 1.68 | 1.75 | 0.52 | 1.48 | 5.16 | 0.97 | 0.82 | 5.16 | 0.97 | 1.60 |
| LoRA + Temp | 1.78 | 0.51 | 1.62 | 1.77 | 0.50 | 1.54 | 5.22 | 0.92 | 0.81 | 5.25 | 0.92 | 1.49 |
| Dropout (S=4) | 1.77 | 0.50 | 1.54 | 1.70 | 0.50 | 1.59 | 5.21 | 0.94 | 0.79 | 5.19 | 0.95 | 1.51 |
| Bayesian-LoRA (S=1) | 1.73 | 0.51 | 1.46 | 1.71 | 0.51 | 1.51 | 5.14 | 0.95 | 0.80 | 5.14 | 0.92 | 1.31 |
| Bayesian-LoRA (S=2) | **1.70** | **0.48** | **1.36** | **1.66** | **0.48** | **1.41** | **5.13** | **0.91** | **0.79** | **5.12** | **0.90** | **1.26** |

vides clear calibration benefits with notably lower NLL and ECE compared to standard fine-tuning, even at the 14B and 30B scales.

## 4.3 EFFICIENCY

In order to present the efficiency of the computational footprint of **Bayesian-LoRA**, we empirically report the number of parameters, training time (per epoch), peak GPU memory usage during training, per-batch inference latency, and sample settings in comparison with the baselines. Bayesian-LoRA introduces only $\mathcal{O}(rc)$ additional negligible parameters, where $r$ and $c$ denote the shape of the inducing matrix, for the inducing variables and kernel factors. Peak GPU memory refers to the maximum memory requirement for training. Per-batch inference latency denotes the inference cost; here, we use a batch size of 5. The sample setting indicates how many samples are taken to approximate the predictive distribution score. All comparison metrics are normalized by setting MAP (LoRA) to 1, i.e., each value is reported as a multiple relative to MAP.

Table 4: Efficiency comparison normalized to standard LoRA (MAP) on the WinoGrande-M dataset. All results of the comparison methods are reproduced based on open-source code.

| Method | Trainable Parameters | Train time (×MAP) | Peak memory (×MAP) | Inference latency (×MAP) | Samples (Val) |
|---|---|---|---|---|---|
| MAP (LoRA) (Hu et al., 2022) | 4.48M | 1.00 | 1.00 | 1.00 | 1 |
| Dropout (Gal & Ghahramani, 2016) | 4.48M | $\approx 4\times$ | $\approx 1\times$ | $\approx 4\times$ | 4 |
| Ckpt Ens (3) (Huang et al., 2017) | $1\times$ MAP | $\approx 1\times$ | $\approx 1\times$ | $\approx 3\times$ | 3 |
| Deep Ens (3) (Lakshminarayanan et al., 2017) | $3\times$ MAP | $\approx 3\times$ | $\approx 3\times$ | $\approx 3\times$ | 3 |
| **BBB** (Blundell et al., 2015) | $\approx 2\times$ MAP | $\approx 4.19\times$ | $\approx 1.05\times$ | $\approx 4.6\times$ | 4 |
| BLoB (N=10) (Wang et al., 2024a) | $\approx 1.5\times$MAP | $\approx 1.11\times$ | $\approx 0.95\times$ | $\approx 6.3\times$ | 4 |
| LLLA (post-hoc) (Yang et al., 2024) | 4.48M + KFAC ($\approx 0.36$M) | $\approx 1.052\times$ | $\approx 1.002\times$ | $\approx 1.9\times$ | - |
| LA (post-hoc) (Yang et al., 2024) | 4.48M + KFAC ($\approx 4.98$M) | $\approx 1.117\times$ | $\approx 1.004\times$ | $\approx 4.36\times$ | - |
| Bayesian-LoRA (End-to-End) | 4.9M | $\approx 1.229\times$ | $\approx 1.003\times$ | $\approx 1.516\times$ / $\approx 2.790\times$ | 2 / 4 |

In Table 4, we show a comparison of the different methods in terms of parameter size, training, and inference overhead (all normalized relative to MAP). This shows that ensemble methods (Huang et al., 2017) significantly increase the number of parameters and inference latency; Bayesian methods, such as BBB (Blundell et al., 2015) and BLoB (Wang et al., 2024a), increase training and inference costs to some extent, especially BLoB, which incurs significant expenses during the inference phase. Although post-hoc methods such as LA and LLLA have relatively controllable parameter quantities, the inference latency also increases significantly. In contrast, Bayesian-LoRA introduces only a very small number of additional parameters, and the training time and memory usage are

nearly the same as those of standard LoRA. The additional cost in the inference stage mainly depends on the number of samples, and settings of 2 or 4 are typically effective. Under these settings, **Bayesian-LoRA** remains close to MAP in efficiency.

### 4.4 ABLATION

In order to systematically analyze the role of the flow layer in the posterior transform on the model, we conducted ablation experiments on the OBQA dataset. In Table 5, when L=0 (i.e., without using flow and relying solely on sparse Gaussian process priors), the accuracy and calibration performance of the model show some decrease. As $L$ increases from 0 to 1, accuracy, ECE, and NLL all improve, but the computational overhead also grows. The single-layer flow ($L = 1$) offers a favorable trade-off between performance gains and efficiency, and we therefore adopt it as our default setting.

Although the deeper flow layer has shown slight improvements in some indicators, it has not shown significant advantages when considering both performance and cost. As mentioned in Table 1, we chose the number of inducing points as 9 to maintain the same number of (trainable) parameters for a fair comparison. Furthermore, we further conducted ablation experiments on the inducing points in **Bayesian-LoRA**, as shown in Figure 2. We compare the differences of parameters in Appendix H, Table **??**.

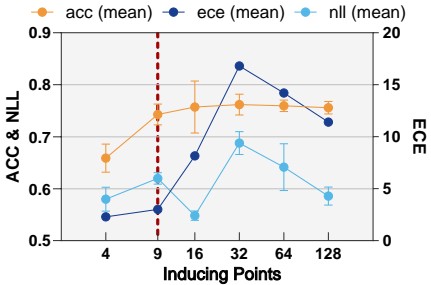

Figure 2: Ablation on inducing points $r = c$

Table 5: Ablation on flow depth $L$ in the posterior transform $T_\phi$ on OBQA. Values are macro-averages (mean $\pm$ std over 3 seeds). $\Delta$ = current $-$ value at $L$=1. ACC/ECE are in percentage points; NLL is an absolute difference. Efficiency is relative to standard LoRA (MAP).

| Flow depth $L$ | ACC ↑ | | ECE ↓ | | NLL ↓ | | Efficiency ($\times$ MAP) | |
|---|---|---|---|---|---|---|---|---|
| | value | $\Delta$ vs. $L$=1 | value | $\Delta$ vs. $L$=1 | value | $\Delta$ vs. $L$=1 | train time | peak mem. |
| 0 (pure SGP) | $79.0 \pm 0.21$ | -2.6 | $5.8 \pm 0.13$ | +0.1 | $0.58 \pm 0.08$ | +0.01 | 1.19 | 1.002 |
| 1 | $81.6 \pm 0.10$ | 0.0 | $5.7 \pm 0.20$ | 0.0 | $0.57 \pm 0.12$ | 0.00 | 1.23 | 1.003 |
| 2 | $80.8 \pm 0.14$ | -0.8 | $5.6 \pm 0.09$ | -0.1 | $0.52 \pm 0.06$ | -0.05 | 1.30 | 1.008 |
| 4 | $80.9 \pm 0.08$ | -0.7 | $4.9 \pm 0.03$ | -0.8 | $0.48 \pm 0.13$ | -0.09 | 1.38 | 1.010 |

### 4.5 ROBUSTNESS EVALUATION

To evaluate the robustness of the proposed method in out-of-distribution (OOD) scenarios, we conducted comparative experiments on six OOD datasets, covering two types of distribution offsets: relatively small offsets (ARC-C, ARC-E) and larger offsets (CS, Eng, Law, Health) from the MMLU dataset. For details, see Appendix G.

From Table 6, we can see that the post-processing methods such as LA/LLLA demonstrate significant calibration advantages (lower ECE) under ID and smaller offset conditions, but such improvements do not always translate into consistent accuracy improvements; Integrating with Bayesian baselines (such as Ckpt Ens, BLoB, BBB) brings more robust likelihood performance (lower NLL) on several datasets, at the cost of higher inference or training costs.

Compared with the above methods, **Bayesian-LoRA** has an overall advantage in NLL metrics in larger offset scenarios, and maintains accuracy comparable to the optimal results and/or better calibration on multiple datasets simultaneously; Based on the efficiency results presented in the previous section (see Table 4), **Bayesian-LoRA** achieves OOD performance comparable to or better than heavy ensembles and large ensembles while maintaining low additional computational costs.

### 4.6 CONSTRAINED BAYESIAN HYPERPARAMETER OPTIMIZATION FOR BAYESIAN-LORA

In Table 1, in order to ensure fairness in the comparison, we adopted the same hyperparameter settings as Yang et al. (2024). However, these hyperparameters may not necessarily be the most suitable

Table 6: Robustness Comparison of different methods across six out-of-distribution datasets that include two types of shift, small and large. Experimental settings follow (Yang et al., 2024).

| Metrics | Methods | ID OBQA | Smaller Distribution Shift ARC-C | ARC-E | CS | Larger Distribution Shift Eng | Law | Health |
|---|---|---|---|---|---|---|---|---|
| ACC ↑ | MAP (Hu et al., 2022) | $78.7 \pm 0.4$ | $67.9 \pm 1.4$ | $77.7 \pm 0.3$ | $42.0 \pm 3.2$ | $41.2 \pm 2.0$ | $37.4 \pm 0.4$ | $48.3 \pm 0.3$ |
| | Dropout (Gal & Ghahramani, 2016) | $79.5 \pm 0.2$ | $67.7 \pm 0.6$ | $77.2 \pm 0.6$ | $41.9 \pm 2.2$ | $39.6 \pm 1.7$ | $37.9 \pm 0.4$ | $48.2 \pm 0.9$ |
| | Ckpt Ens (Huang et al., 2017) | $79.1 \pm 0.2$ | $67.9 \pm 0.8$ | $77.4 \pm 0.8$ | $41.1 \pm 2.0$ | $38.7 \pm 1.2$ | $37.7 \pm 0.2$ | $48.2 \pm 0.6$ |
| | Temp (Guo et al., 2017) | $77.7 \pm 0.8$ | $68.0 \pm 0.2$ | $76.7 \pm 1.0$ | $43.5 \pm 0.9$ | $\mathbf{44.4 \pm 2.0}$ | $37.4 \pm 0.1$ | $47.7 \pm 0.8$ |
| | BBB (Blundell et al., 2015) | $77.0 \pm 0.2$ | $67.3 \pm 1.2$ | $75.8 \pm 0.8$ | $40.5 \pm 0.3$ | $36.7 \pm 0.2$ | $36.1 \pm 0.2$ | $47.6 \pm 0.5$ |
| | BLoB(N=10) (Wang et al., 2024a) | $81.5 \pm 0.7$ | $67.7 \pm 1.1$ | $76.3 \pm 0.8$ | $44.6 \pm 0.4$ | $42.3 \pm 1.7$ | $37.4 \pm 1.2$ | $47.3 \pm 1.5$ |
| | LLLA (Yang et al., 2024) | $78.7 \pm 0.4$ | $68.1 \pm 0.0$ | $78.1 \pm 0.0$ | $45.6 \pm 0.0$ | $38.9 \pm 0.0$ | $37.1 \pm 0.0$ | $48.5 \pm 0.0$ |
| | LA (Yang et al., 2024) | $78.9 \pm 0.4$ | $69.2 \pm 0.0$ | $78.5 \pm 0.0$ | $45.1 \pm 0.0$ | $39.1 \pm 0.0$ | $37.3 \pm 0.0$ | $49.1 \pm 0.0$ |
| | Bayesian-LoRA (S=4) (End-to-End) | $\mathbf{81.6 \pm 0.1}$ | $\mathbf{69.5 \pm 0.1}$ | $\mathbf{78.9 \pm 0.2}$ | $\mathbf{46.3 \pm 0.1}$ | $37.5 \pm 0.2$ | $\mathbf{37.9 \pm 0.1}$ | $\mathbf{50.6 \pm 0.1}$ |
| ECE ↓ | MAP (Hu et al., 2022) | $16.1 \pm 0.6$ | $22.2 \pm 1.2$ | $15.8 \pm 1.0$ | $34.2 \pm 3.1$ | $38.4 \pm 1.7$ | $35.2 \pm 0.7$ | $34.2 \pm 0.8$ |
| | Dropout (Gal & Ghahramani, 2016) | $15.0 \pm 0.4$ | $21.4 \pm 0.4$ | $15.5 \pm 1.0$ | $33.7 \pm 2.0$ | $38.6 \pm 2.9$ | $34.2 \pm 0.6$ | $33.5 \pm 0.2$ |
| | Ckpt Ens (Huang et al., 2017) | $10.1 \pm 0.3$ | $17.7 \pm 0.7$ | $12.1 \pm 0.6$ | $29.1 \pm 2.3$ | $32.5 \pm 1.8$ | $32.1 \pm 0.1$ | $29.0 \pm 0.3$ |
| | Temp (Guo et al., 2017) | $7.20 \pm 2.6$ | $9.41 \pm 3.5$ | $6.33 \pm 1.3$ | $16.4 \pm 5.5$ | $16.1 \pm 4.6$ | $22.7 \pm 5.8$ | $17.4 \pm 6.0$ |
| | BBB Blundell et al. (2015) | $11.3 \pm 1.1$ | $19.9 \pm 0.7$ | $13.4 \pm 0.9$ | $18.6 \pm 1.6$ | $22.4 \pm 3.1$ | $28.1 \pm 2.5$ | $27.4 \pm 1.8$ |
| | BLoB (N=10) Wang et al. (2024a) | $3.77 \pm 1.5$ | $9.55 \pm 0.4$ | $5.48 \pm 1.3$ | $12.6 \pm 1.7$ | $22.3 \pm 1.9$ | $25.3 \pm 2.1$ | $16.4 \pm 1.6$ |
| | LLLA (Yang et al., 2024) | $15.8 \pm 0.6$ | $21.3 \pm 0.0$ | $14.8 \pm 0.0$ | $30.3 \pm 0.0$ | $39.7 \pm 0.0$ | $33.6 \pm 0.0$ | $33.5 \pm 0.0$ |
| | LA (Yang et al., 2024) | $\mathbf{3.50 \pm 0.4}$ | $\mathbf{5.50 \pm 0.0}$ | $\mathbf{3.10 \pm 0.0}$ | $14.5 \pm 0.0$ | $\mathbf{12.8 \pm 0.0}$ | $23.9 \pm 0.0$ | $17.6 \pm 0.0$ |
| | Bayesian-LoRA (S = 4) (End-to-End) | $5.70 \pm 0.2$ | $8.10 \pm 0.1$ | $5.21 \pm 0.1$ | $\mathbf{11.1 \pm 0.0}$ | $20.4 \pm 0.1$ | $\mathbf{16.5 \pm 0.0}$ | $\mathbf{12.9 \pm 0.0}$ |
| NLL ↓ | MAP (Hu et al., 2022) | $0.99 \pm 0.05$ | $1.30 \pm 0.07$ | $1.04 \pm 0.10$ | $1.90 \pm 0.12$ | $2.19 \pm 0.15$ | $2.12 \pm 0.03$ | $2.09 \pm 0.08$ |
| | Dropout (Gal & Ghahramani, 2016) | $0.95 \pm 0.04$ | $1.24 \pm 0.06$ | $1.01 \pm 0.09$ | $1.86 \pm 0.10$ | $2.14 \pm 0.13$ | $2.09 \pm 0.02$ | $2.05 \pm 0.07$ |
| | Ckpt Ens (Huang et al., 2017) | $0.68 \pm 0.03$ | $1.03 \pm 0.03$ | $0.80 \pm 0.03$ | $1.55 \pm 0.04$ | $1.72 \pm 0.01$ | $1.94 \pm 0.01$ | $1.74 \pm 0.02$ |
| | Temp (Guo et al., 2017) | $0.67 \pm 0.02$ | $0.90 \pm 0.05$ | $0.66 \pm 0.01$ | $1.31 \pm 0.06$ | $1.32 \pm 0.07$ | $1.65 \pm 0.16$ | $1.36 \pm 0.10$ |
| | BBB (Blundell et al., 2015) | $0.66 \pm 0.05$ | $1.06 \pm 0.01$ | $0.79 \pm 0.02$ | $1.49 \pm 0.05$ | $1.83 \pm 0.08$ | $1.65 \pm 0.20$ | $1.29 \pm 0.12$ |
| | BLoB (N=10) (Wang et al., 2024a) | $0.50 \pm 0.01$ | $0.83 \pm 0.01$ | $\mathbf{0.60 \pm 0.01}$ | $1.38 \pm 0.01$ | $1.81 \pm 0.10$ | $2.01 \pm 0.09$ | $1.94 \pm 0.08$ |
| | LLLA (Yang et al., 2024) | $0.66 \pm 0.02$ | $0.88 \pm 0.00$ | $0.64 \pm 0.00$ | $1.28 \pm 0.00$ | $1.27 \pm 0.00$ | $1.49 \pm 0.00$ | $1.31 \pm 0.00$ |
| | LA (Yang et al., 2024) | $0.62 \pm 0.01$ | $0.85 \pm 0.00$ | $0.62 \pm 0.00$ | $1.26 \pm 0.00$ | $1.27 \pm 0.00$ | $1.68 \pm 0.00$ | $1.35 \pm 0.00$ |
| | Bayesian-LoRA (S=4) (End-to-End) | $\mathbf{0.49 \pm 0.02}$ | $\mathbf{0.82 \pm 0.04}$ | $0.80 \pm 0.04$ | $\mathbf{1.22 \pm 0.04}$ | $\mathbf{1.26 \pm 0.02}$ | $1.42 \pm 0.04$ | $1.20 \pm 0.02$ |

for **Bayesian-LoRA**. Hence, we use a constrained Bayesian optimization method to determine hyperparameters that are more suitable for our method.

We tune the learning rate and weight decay, using a multi-objective Gaussian Process (GP) approach. On the training set, we measure calibration, likelihood, and accuracy, and convert accuracy to a minimization objective:

$$\min_{\mathbf{x} \in \mathcal{X}} \mathbf{f}(\mathbf{x}) = \big(f_1(\mathbf{x}), f_2(\mathbf{x}), f_3(\mathbf{x})\big) = \big(\text{ECE, NLL, } -\text{ACC}\big), \qquad (13)$$

We use GP as the surrogate model and maximize the *noisy expected hypervolume improvement* (NEHVI) to obtain Pareto-front improvement (details in Appendix H)

In Figures 3 and 6 and Table 14, we present all configuration details for the learning rate and weight decay. Note that "best choice" in Figures denotes the configuration we adopted in the experiments that achieves a relatively good trade-off between accuracy and uncertainty estimation. The final results are summarized in Table 7, comparing performance before and after BO.

Table 7: Metrics before vs. after constrained Bayesian optimization. "Before" uses Table 1 settings (Bayesian-LoRA, $S=4$, early stop); "After (BO)" are results under constrained BO

| Dataset | ACC ↑ Before | After (BO) | ECE ↓ Before | After (BO) | NLL ↓ Before | After (BO) | Hyperparams (BO) LR (BO) | WD (BO) |
|---|---|---|---|---|---|---|---|---|
| WG-S | 70.90 | **72.94** | 4.90 | **2.74** | 0.79 | **0.54** | $9.792 \times 10^{-5}$ | $9.339 \times 10^{-2}$ |
| ARC-C | 68.90 | **69.60** | 9.20 | **6.10** | **0.77** | 0.86 | $4.955 \times 10^{-4}$ | $2.056 \times 10^{-1}$ |
| ARC-E | 85.91 | **88.38** | 5.30 | **4.97** | **0.38** | 0.39 | $7.393 \times 10^{-4}$ | $2.707 \times 10^{-2}$ |
| WG-M | 74.30 | **76.34** | **3.00** | 7.90 | 0.62 | **0.52** | $4.280 \times 10^{-4}$ | $1.000 \times 10^{-2}$ |
| OBQA | 81.60 | **82.80** | **5.70** | 5.84 | **0.49** | 0.54 | $6.552 \times 10^{-5}$ | $9.712 \times 10^{-2}$ |
| BoolQ | 86.10 | **86.41** | **2.10** | 3.10 | 0.29 | **0.28** | $5.421 \times 10^{-4}$ | $3.582 \times 10^{-1}$ |

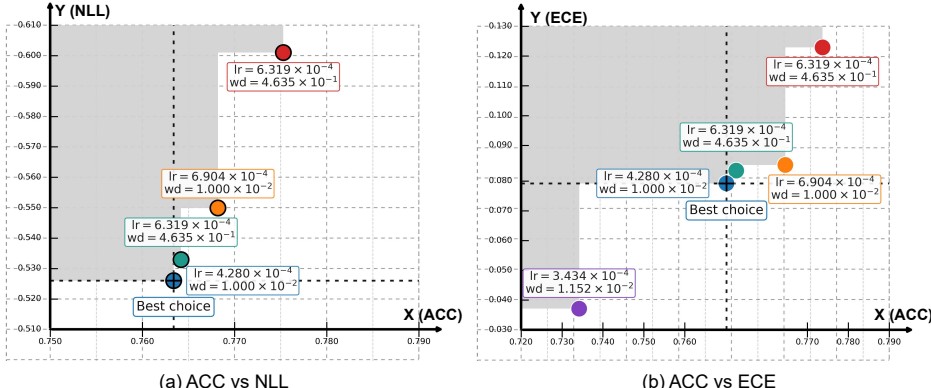

(a) ACC vs NLL          (b) ACC vs ECE

Figure 3: Pareto analysis on the WinoGrande-M dataset (ACC–NLL and ACC–ECE). Each point denotes a hyperparameter pair (lr, wd). Gray regions show dominated solutions, and the cross marks the "Best choice" near the Pareto front.

## 5 CONCLUSION

In this work, we proposed **Bayesian-LoRA**, a framework integrating Sparse Gaussian Process with native LoRA and applying normalizing flow to stabilize the training process. Across six common-sense reasoning benchmarks with a training budget comparable to standard LoRA, Bayesian-LoRA achieves the best or near-best accuracy on most datasets.

## 6 REPRODUCIBILITY STATEMENT

To ensure the reproducibility of the results, we provide the following:

- An anonymous code repository is available at: `https://anonymous.4open.science/r/Bayesian-LoRA`.

- All training and model hyperparameters are listed in Tables 11 and 12.

- The experiments were conducted on PyTorch 2.5.0, HuggingFace Transformers 4.51.3, and CUDA 12.8. We have also clearly listed the required Python dependencies in the **requirements.txt** file for easy installation.

- All datasets used in the experiment are publicly available, including WinoGrande-S (Sakaguchi et al., 2020), ARC-C (Clark et al., 2018), ARC-E (Clark et al., 2018), WinoGrande-M (Sakaguchi et al., 2020), OBQA (Mihaylov et al., 2018), BoolQ (Clark et al., 2019) and WikiText-2 (Merity et al., 2016). We provide scripts for automatic download and preprocessing.

- We fixed random seeds in PyTorch, NumPy, and CUDA to reduce variance. The results are reported in the form of the average value $\pm$ standard deviation of three runs.

- The training was completed on $4 \times$ A100-80GB GPUs. The peak memory, training time, and inference delay have been provided as references in Section 4.3.

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

# A DERIVATIONS FOR THE VARIATIONAL SPARSE INDUCING WEIGHT MODEL

## A.1 MODEL SPECIFICATION AND JOINT DENSITY

We define $W \in \mathbb{R}^{d_{\text{out}} \times d_{\text{in}}}$ being the layer weight matrix and $U \in \mathbb{R}^{r \times c}$ the low-dimensional inducing matrix with $r \ll d_{\text{out}}$ and $c \ll d_{\text{in}}$. We place a Gaussian (equivalently, matrix-normal) prior on $U$:

$$p(U) = \mathcal{N}\big(\text{vec}(U) \mid \mathbf{0}, \mathbf{K}_U\big), \qquad \mathbf{K}_U = \mathbf{K}_c \otimes \mathbf{K}_r, \tag{14}$$

which is equivalent to $U \sim \mathcal{MN}(\mathbf{0}, \mathbf{K}_r, \mathbf{K}_c)$. Given $U$, the conditional distribution of $W$ is Gaussian

$$p(W \mid U) = \mathcal{N}\big(W \mid M_W(U), \lambda^2 \Sigma_W\big), \tag{15}$$

where $\lambda > 0$ is a scale parameter and $M_W(U)$ is linear in $U$:

$$M_W(U) = T_r U T_c, \qquad T_r = Z_r^\top K_r^{-1}, \quad T_c = K_c^{-1} Z_c, \tag{16}$$

with

$$K_r = Z_r Z_r^\top + D_r^2, \qquad K_c = Z_c Z_c^\top + D_c^2. \tag{17}$$

The likelihood factorizes over data $\mathcal{D} = \{(x_n, y_n)\}_{n=1}^N$ as $p(\mathcal{D} \mid W) = \prod_{n=1}^N p(y_n \mid f(x_n; W))$. And the joint density could be described as:

$$p(U, W, \mathcal{D}) = p(U) \, p(W \mid U) \, p(\mathcal{D} \mid W) \tag{18}$$

## A.2 VARIATIONAL FAMILY AND FACTORIZATION

We posit a Gaussian variational posterior on $U$,

$$q(U) = \mathcal{N}\big(\text{vec}(U) \mid \mathbf{m}, \mathbf{S}\big), \tag{19}$$

and keep the model conditional $p(W \mid U)$ inside the variational family:

$$q(U, W) = q(U) \, p(W \mid U). \tag{20}$$

Marginalizing $U$ gives the variational distribution over $W$:

$$q(W) = \int p(W \mid U) \, q(U) \, dU. \tag{21}$$

## A.3 ELBO DERIVATION

Starting from $\log p(\mathcal{D}) = \log \int p(U, W, \mathcal{D}) \, dU \, dW$ and inserting $q(U, W)$:

$$\log p(\mathcal{D}) = \log \int q(U, W) \frac{p(U) \, p(W \mid U) \, p(\mathcal{D} \mid W)}{q(U) \, p(W \mid U)} \, dU \, dW, \tag{22}$$

Jensen's inequality yields the ELBO

$$\mathcal{L} = \mathbb{E}_{q(U)p(W|U)}\Big[ \log p(\mathcal{D} \mid W) \Big] - \mathbb{E}_{q(U)}\Big[ \log \tfrac{q(U)}{p(U)} \Big], \tag{23}$$

i.e.

$$\mathcal{L} = \mathbb{E}_{q(W)}\Big[ \log p(\mathcal{D} \mid W) \Big] - \text{KL}\big(q(U) \, \| \, p(U)\big). \tag{24}$$

Equivalently, by adding and subtracting $\mathbb{E}_{q(U)}[\log p(W \mid U)]$,

$$\mathcal{L} = \mathbb{E}_{q(W)}\Big[ \log p(\mathcal{D} \mid W) \Big] - \text{KL}\big(q(U) \, \| \, p(U)\big) - \mathbb{E}_{q(U)}\Big[ \text{KL}\big(q(W \mid U) \, \| \, p(W \mid U)\big) \Big], \tag{25}$$

which matches the presentation in the main text.

### A.4 CLOSED-FORM TERMS

**KL between Gaussians for $q(U)$ and $p(U)$.** Define $d_U = rc$ and denote $\mathbf{m} \in \mathbb{R}^{d_U}$, $\mathbf{S} \in \mathbb{R}^{d_U \times d_U}$, and $\mathbf{K}_U = \mathbf{K}_c \otimes \mathbf{K}_r$. Then

$$\mathrm{KL}\big(q(U) \,\|\, p(U)\big) = \frac{1}{2}\Big(\mathrm{tr}(\mathbf{K}_U^{-1}\mathbf{S}) + \mathbf{m}^\top \mathbf{K}_U^{-1}\mathbf{m} - d_U + \log \tfrac{|\mathbf{K}_U|}{|\mathbf{S}|}\Big). \tag{26}$$

Using $\log |\mathbf{K}_c \otimes \mathbf{K}_r| = c \log |\mathbf{K}_r| + r \log |\mathbf{K}_c|$ simplifies evaluation. In the whitened case with $p(U) = \mathcal{N}(\mathbf{0}, \mathbf{I}_{d_U})$,

$$\mathrm{KL}\big(q(U) \,\|\, p(U)\big) = \frac{1}{2}\Big(\mathrm{tr}(\mathbf{S}) + \mathbf{m}^\top \mathbf{m} - d_U - \log |\mathbf{S}|\Big). \tag{27}$$

**Conditional KL for $q(W \mid U)$ vs $p(W \mid U)$.** Assume both are Gaussians sharing the same mean $M_W(U)$ and with covariances $\Sigma_q = \lambda^2 \Sigma_W$ and $\Sigma_p = \Sigma_W$. Set $d_W = d_{\mathrm{out}}d_{\mathrm{in}}$. Then

$$\mathrm{KL}\big(q(W \mid U) \,\|\, p(W \mid U)\big) = \frac{1}{2}\Big(\mathrm{tr}(\Sigma_p^{-1}\Sigma_q) - d_W + \log \tfrac{|\Sigma_p|}{|\Sigma_q|}\Big) = \frac{d_W}{2}\big(\lambda^2 - 1\big) - d_W \log \lambda. \tag{28}$$

This is independent of $U$ and thus the expectation $\mathbb{E}_{q(U)}[\cdot]$ in equation 25 is trivial.

### A.5 FORM OF $q(W)$: MEAN AND COVARIANCE

Vectorizing $W$ and using $\mathrm{vec}(T_r U T_c) = (T_c^\top \otimes T_r)\,\mathrm{vec}(U)$, one obtains

$$\mathbb{E}_q[\,\mathrm{vec}(W)\,] = (T_c^\top \otimes T_r)\,\mathbf{m}, \qquad \mathbb{E}_q[\,W\,] = T_r\,M\,T_c, \tag{29}$$

where $M$ is the matricized version of $\mathbf{m}$. Moreover, since $\Sigma_q(W \mid U) = \lambda^2 \Sigma_W$ is independent of $U$,

$$\mathrm{Cov}_q[\,\mathrm{vec}(W)\,] = \lambda^2 \Sigma_W + (T_c^\top \otimes T_r)\,\mathbf{S}\,(T_c \otimes T_r^\top). \tag{30}$$

Thus $q(W)$ is Gaussian with the above mean and covariance whenever $q(U)$ is Gaussian and $M_W(U)$ is linear in $U$.

Finally,

$$\log p(\mathcal{D}) = \mathcal{L} + \mathrm{KL}\big(q(U) \,\|\, p(U \mid \mathcal{D})\big), \tag{31}$$

So maximizing $\mathcal{L}$ minimizes the divergence from the variational posterior to the exact posterior over $U$.

**Proposition A.1** (Details of $U$-space-independent KL). *Let $U \in \mathbb{R}^m$ denote the inducing variables with a Gaussian prior $p(U) = \mathcal{N}(\mu_p, \Sigma_p)$ and variational posterior $q_\psi(U) = \mathcal{N}(\mu_q, \Sigma_q)$. Let $T_\phi : \mathbb{R}^m \to \mathbb{R}^m$ be an invertible $C^1$ map (the adapter flow) with Jacobian $J_{T_\phi}(U)$, and define the LoRA parameters as the deterministic pushforward $\Delta W = T_\phi(U)$. For data $\mathcal{D}$ with likelihood $p(\mathcal{D} \mid \Delta W)$, the ELBO in $U$-space*

$$\mathcal{L}(\phi, \psi) = \mathbb{E}_{q_\psi(U)}\big[\log p(\mathcal{D} \mid T_\phi(U))\big] - \mathrm{KL}\big(q_\psi(U) \,\|\, p(U)\big) \tag{32}$$

*is equivalent to the ELBO in $\Delta W$-space,*

$$\mathcal{L}(\phi, \psi) = \mathbb{E}_{q_\phi(\Delta W)}\big[\log p(\mathcal{D} \mid \Delta W)\big] - \mathrm{KL}\big(q_\phi(\Delta W) \,\|\, p_\phi(\Delta W)\big), \tag{33}$$

*where $q_\phi(\Delta W) := T_{\phi\#}q_\psi$ and $p_\phi(\Delta W) := T_{\phi\#}p$ are pushforwards under $T_\phi$. Moreover, the KL term is invariant under $T_\phi$:*

$$\mathrm{KL}\big(q_\phi(\Delta W) \,\|\, p_\phi(\Delta W)\big) = \mathrm{KL}\big(q_\psi(U) \,\|\, p(U)\big). \tag{34}$$

*In particular, when $q_\psi$ and $p$ are Gaussian, the KL admits a closed form*

$$\mathrm{KL}\big(\mathcal{N}(\mu_q, \Sigma_q) \,\|\, \mathcal{N}(\mu_p, \Sigma_p)\big) =$$

$$\tfrac{1}{2}\Big(\mathrm{tr}(\Sigma_p^{-1}\Sigma_q) + (\mu_p - \mu_q)^\top \Sigma_p^{-1}(\mu_p - \mu_q) - m + \log \frac{\det \Sigma_p}{\det \Sigma_q}\Big) \tag{35}$$

*Proof.* Since $\Delta W = T_\phi(U)$ is a deterministic, invertible change of variables, The data term satisfies $\mathbb{E}_{q_\psi(U)}[\log p(\mathcal{D} \mid T_\phi(U))] = \mathbb{E}_{q_\phi(\Delta W)}[\log p(\mathcal{D} \mid \Delta W)]$. For the KL term, write the pushforward densities via change of variables: $q_\phi(\Delta W) = q_\psi(U) |\det J_{T_\phi}(U)|^{-1}$ and $p_\phi(\Delta W) = p(U) |\det J_{T_\phi}(U)|^{-1}$ with $\Delta W = T_\phi(U)$. Then

$$\mathrm{KL}(q_\phi \| p_\phi) = \int q_\phi(\Delta W) \log \frac{q_\phi(\Delta W)}{p_\phi(\Delta W)} \, d\Delta W = \int q_\psi(U) \log \frac{q_\psi(U)}{p(U)} \, dU = \mathrm{KL}(q_\psi \| p), \quad (36)$$

where the Jacobian determinants cancel exactly. Combining the two parts yields the equivalent ELBO forms equation 32–equation 33 and the invariance equation 34; the KL does not depend on $\Delta W$ nor on $T_\phi$. When $q_\psi$ and $p$ are Gaussian, equation 35 follows from the standard closed-form KL between multivariate Gaussians.

## B ALTERNATIVE BAYESIAN-LORA PARAMETERIZATIONS

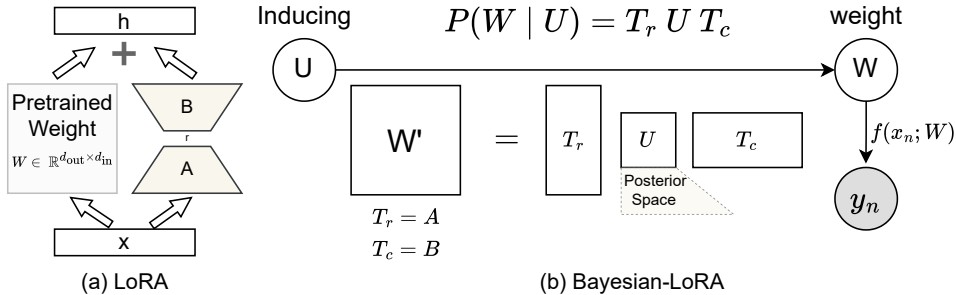

Figure 4: Version-2 (weight-space). The posterior is defined on the weight $W$ via an inducing matrix $U$ with transforms $T_r, T_c$.

We have also provided an alternative implementation of Bayesian LoRA, where the weight $W$ is generated via an inducing matrix $U$ with transforms $T_r, T_c$, which perfectly aligns with the low-rank concept. Compared with Fig. 1, only one small inducing trainable matrix is needed, instead of four such matrices.

## C EFFECT OF MONTE CARLO SAMPLES ON ACCURACY AND UNCERTAINTY

In this section we analyse how the number of Monte Carlo samples $S$ affects predictive accuracy and uncertainty for the in-distribution (ID) ARC dataset and the out-of-distribution (OOD) OBQA dataset at a fixed checkpoint. For each $S$, we report accuracy (ACC), negative log-likelihood (NLL), expected calibration error (ECE; 15 bins), and the average per-batch inference time. Figure 5 shows the relative changes in ACC, NLL, ECE and inference time with respect to $S = 1$ for both ARC (ID) and OBQA (OOD). As $S$ increases, the inference time grows almost linearly and reaches close to an order-of-magnitude increase between $S = 1$ and $S = 10$. On the ARC (ID) data, ACC, NLL and ECE vary only within a very narrow range once $S \geq 2$, and mostly fluctuate rather than improving systematically. On the OBQA (OOD) data, larger $S$ yields slightly lower NLL/ECE and marginally more stable accuracy, consistent with reduced Monte Carlo noise in the predictive distribution. However, the gains beyond $S = 2$–4 remain small compared to the additional latency, which supports our practical recommendation of using $S = 2$–4 as a good trade-off between uncertainty quality and computational cost.

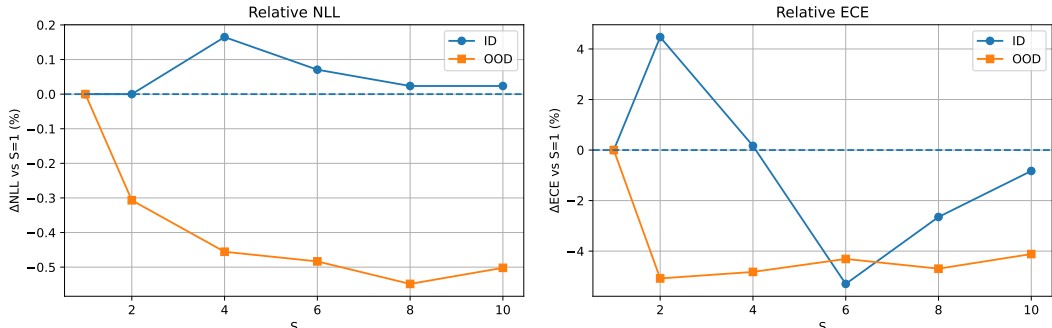

Figure 5: Relative changes in ACC, NLL, ECE and inference time with respect to $S = 1$ for the ID ARC dataset and the OOD OBQA dataset at a fixed checkpoint.

Table 8: Effect of the number of Monte Carlo samples $S$ on OOD performance on OBQA at a fixed checkpoint. Time denotes average per-batch inference time in seconds.

| $S$ | NLL ↓ | ACC ↑ | ECE ↓ | Time (s) ↓ |
|---|---|---|---|---|
| 1 | 1.0756 | 0.6333 | 0.1555 | 2.3997 |
| 2 | 1.0723 | 0.6354 | 0.1476 | 4.7858 |
| 3 | 1.0716 | 0.6333 | 0.1516 | 7.1729 |
| 4 | 1.0707 | 0.6354 | 0.1480 | 9.5596 |
| 5 | 1.0708 | 0.6313 | 0.1521 | 11.9460 |
| 6 | 1.0704 | 0.6354 | 0.1488 | 14.3300 |
| 7 | 1.0699 | 0.6333 | 0.1503 | 16.7140 |
| 8 | 1.0697 | 0.6354 | 0.1482 | 19.0961 |
| 9 | 1.0697 | 0.6354 | 0.1509 | 21.4805 |
| 10 | 1.0702 | 0.6354 | 0.1491 | 23.8661 |

Table 9: Effect of the number of Monte Carlo samples $S$ on ID performance on ARC at the same checkpoint. Time denotes average per-batch inference time in seconds.

| $S$ | NLL ↓ | ACC ↑ | ECE ↓ | Time (s) ↓ |
|---|---|---|---|---|
| 1 | 0.4245 | 0.8697 | 0.0604 | 0.7610 |
| 2 | 0.4245 | 0.8662 | 0.0631 | 1.5253 |
| 3 | 0.4253 | 0.8662 | 0.0644 | 2.2888 |
| 4 | 0.4252 | 0.8662 | 0.0605 | 3.0507 |
| 5 | 0.4252 | 0.8680 | 0.0584 | 3.8133 |
| 6 | 0.4248 | 0.8680 | 0.0572 | 4.5761 |
| 7 | 0.4246 | 0.8662 | 0.0606 | 5.3384 |
| 8 | 0.4246 | 0.8680 | 0.0588 | 6.1010 |
| 9 | 0.4249 | 0.8680 | 0.0572 | 6.8632 |
| 10 | 0.4246 | 0.8680 | 0.0599 | 7.6256 |

## D DATASETS AND PREPROCESSING

This appendix details the six benchmarks used in our evaluation, together with the scoring rules, prompting templates, and implementation choices shared across datasets.

### D.1 TASK OVERVIEW

All tasks are cast as *closed-set prediction* to avoid generation artifacts. For multiple-choice datasets (ARC-C/E, OBQA) and cloze-style coreference (WinoGrande S/M), we compute option probabilities without free decoding; for BoolQ, we map to a binary verbalizer. Table 10 summarizes formats.

Table 10: Task formats and primary metrics.

| Dataset | Task type | Candidates | Primary metric(s) |
|---|---|---|---|
| WinoGrande-S/M (WG-S/M) | Cloze coreference (2-way) | 2 | ACC, ECE, NLL |
| ARC-Challenge (ARC-C) | Multi-choice science QA | 3–5 (mostly 4) | ACC, ECE, NLL |
| ARC-Easy (ARC-E) | Multi-choice science QA | 3–5 (mostly 4) | ACC, ECE, NLL |
| OpenBookQA (OBQA) | Multi-choice science QA | 4 | ACC, ECE, NLL |
| BoolQ | Yes/No reading comprehension | 2 | ACC, ECE, NLL |

**Common evaluation protocol.** Unless otherwise noted: (i) we use the official training/validation/test splits; (ii) Accuracy (ACC ↑), Expected Calibration Error (ECE ↓, 15 bins on $[0, 1]$ using the model's predicted probability for the chosen label), and Negative Log-Likelihood (NLL ↓) are reported; (iii) probabilities are computed from normalized option log-likelihoods as detailed below; (iv) casing and punctuation are preserved.

**Tokenizer and limits.** We use the backbone's SentencePiece tokenizer. Inputs are truncated to a maximum of $L_{\max}$ tokens (default 1024) by trimming long contexts first while keeping all answer options intact. Padding is on the right; the BOS/EOS usage follows the backbone defaults.

**Scoring rule for multiple-choice/cloze.** Given a prefix prompt $x$ and an option string $y_j$ tokenized as $(y_{j,1}, \ldots, y_{j,T_j})$, we score

$$s_j = \frac{1}{T_j} \sum_{t=1}^{T_j} \log p_\theta(y_{j,t} \mid x, y_{j,<t}),$$

i.e., *length-normalized* log-likelihood to mitigate option-length bias. Predicted label $\hat{y} = \arg\max_j s_j$; option probabilities are $\pi_j \propto \exp(s_j)$ and are used for ECE/NLL. NLL is $-\log \pi_{y^\star}$ for gold label $y^\star$. In case of ties, we choose the option with the larger unnormalized (sum) log-likelihood, then lexicographically.

D.2 PROMPTING TEMPLATES

To minimize prompt sensitivity, we use deterministic templates without instructions or chain-of-thought. Placeholders are written as <...>.

**WinoGrande (S/M).** Each instance contains a sentence with a blank and two candidate fillers.

```
Sentence: <sentence>
Option A: <sentence with option1>
Option B: <sentence with option2>
```

We compare the completion likelihoods as in the scoring rule.

**ARC-C / ARC-E.**

```
Question: <question>
Options:
A. <choice A>
B. <choice B>
C. <choice C>
D. <choice D> [E. <choice E> if present]
Answer:
```

Each option is scored by appending its text after `Answer:`.

**OpenBookQA (OBQA).** Same as ARC; four options (A–D). We omit the provided "facts" in the main results for parity.

**BoolQ.**   Binary QA with verbalizers `Yes`/`No`.

```
Passage:   <passage>
Question:  <question>
Answer:
```

We score `Yes` and `No` as the two candidates.

### D.3   PREPROCESSING AND NORMALIZATION

- **Unicode/whitespace.** Normalize Unicode quotes/dashes; strip leading/trailing whitespace; collapse repeated spaces inside fields without altering semantics.

- **Deduplication.** Remove exact duplicate training examples (rare in these corpora); evaluation splits remain untouched.

- **Invalid items.** For ARC items with empty or malformed options, we drop the instance *from training only* and keep evaluation intact; such cases are logged ($< 0.1\%$ in our runs).

- **Context truncation.** When sequences exceed $L_{\max}$, we retain the full question and all options and truncate supporting passages from the left (BoolQ) or auxiliary fields (if used), preserving the end of the passage, which often contains the answer evidence.

- **Label integrity.** All label letters (A/B/...) are taken from the official annotations; we do not remap or reorder options.

### D.4   DATASET-SPECIFIC NOTES

**WinoGrande S/M.**   We use the official S and M partitions. Following common practice, we evaluate on the validation set for early stopping and report test numbers using the official held-out split when available. No external coreference resources are used.

**ARC-Challenge / ARC-Easy.**   We use the AI2 ARC v1 format. Some questions have three or five options; our scoring rule handles variable cardinality. We do not incorporate retrieval or external knowledge in the main comparison.

**OpenBookQA.**   We use the main OBQA v1.1 multiple-choice set. The "open book" facts are omitted in the main results for parity; adding them can increase accuracy, but does not change the calibration trends observed.

**BoolQ.**   We keep case and punctuation in passages. Extremely long passages are truncated from the left to respect $L_{\max}$ while keeping the full question. Verbalizers are fixed as `Yes`/`No`; alternative synonyms (e.g., `True`/`False`) yield similar results.

## E   METRICS FOR UNCERTAINTY QUANTIFICATION

NLL and ECE are two common metrics for quantifying model uncertainty.

**Expected Calibration Error (ECE).**   We compute ECE over $B = 15$ equal-width bins on $[0, 1]$ for the predicted confidences $c_i$. Let $I_b = ((b-1)/B, \, b/B]$ and $B_b = \{ \, i : c_i \in I_b \, \}$. Then:

$$\text{ECE} = \sum_{b=1}^{B} \frac{|B_b|}{N} \left| \text{acc}(B_b) - \text{conf}(B_b) \right|. \tag{37}$$

$$\text{acc}(B_b) = \frac{1}{|B_b|} \sum_{i \in B_b} \mathbf{1}\{\hat{y}_i = y_i\}. \tag{38}$$

$$\text{conf}(B_b) = \frac{1}{|B_b|} \sum_{i \in B_b} c_i, \qquad c_i = \max_k p_\theta(y{=}k \mid x_i). \tag{39}$$

**Negative Log-Likelihood (NLL).** For instance $i$ with gold label $y_i$ and prediction of $\hat{y}_i$

$$\text{NLL} = -\frac{1}{N} \sum_i \log P(\hat{y}_i = y_i) \tag{40}$$

For datasets with variable option counts, $\hat{y}_i$ is always the softmax of length-normalized scores across the *present* options.

## F   HYPERPARAMETERS

Hyperparameters are of crucial importance for reproducibility, and therefore, we have listed all used hyperparameters in **Bayesian-LoRA**.

Table 11: Hyperparameters used in our experiments.

| Hyper-parameter | Value |
|---|---|
| Optimizer | AdamW |
| Learning rate | $5 \times 10^{-4}$ |
| Betas | (0.9, 0.999) |
| Epsilon ($\epsilon$) | $1 \times 10^{-5}$ |
| Weight decay | 0.1 |
| Scheduler | MultiStepLR (milestones = [4, 6], $\gamma = 0.1$) |
| Epochs | 10 |
| Batch size (train) | cfg.dset.train_bs |
| Batch size (eval) | cfg.dset.eval_bs |
| MC samples ($n_{mc}$) | 2 (default, can be set in cfg.eval) |
| KL scaling | 0.2/steps_per_epoch |
| Evaluation frequency | Every 2 epochs |
| Label smoothing | 0.05 |

Table 11 lists all hyperparameters. We have also provided additional adjustable hyperparameters in the configuration that provide flexibility for reproducibility.

Also, the core settings of the Sparse Gaussian Process are listed in Table 12. Inducing rows and columns are set to 9, corresponding to the same parameter level as LoRA. The Q and K matrices of attention and the output linear layers are replaced by the Bayesian framework (SGP).

## G   OUT OF DISTRIBUTION DATASET

Following (Yang et al., 2024), we adapted the Computer Science (CS), Engineering (Eng), Law, and Health subsets of the MMLU dataset as out-of-distribution data to evaluate the robustness of **Bayesian-LoRA**. Table 13 summarizes the MMLU subjects used for OOD evaluation. Following the HuggingFace description of the MMLU dataset (Hendrycks et al., 2021), we split college computer science, computer security, high school computer science, and machine learning into the CS category, electrical engineering into the Eng category, international law, jurisprudence, and professional law into the Law category, and anatomy, clinical knowledge, college medicine, human aging, nutrition, professional medicine, and virology into the Health category.

Table 12: Inducing hyper-parameters.

| Hyper-parameter | Value |
|---|---|
| inducing_rows | 9 |
| inducing_cols | 9 |
| whitened_u | True |
| q_inducing | diagonal |
| learn_lambda | True |
| init_lambda | 0.001 |
| max_lambda | 0.03 |
| max_sd_u | 0.1 |
| cache_cholesky | True |
| prior_sd | 0.1 |
| sqrt_width_scaling | True |
| key_layers | {q_proj, k_proj, lm_head} |

| Subject | Tasks |
|---|---|
| Computer Science (CS) | college computer science, computer security, high school computer science, machine learning |
| Engineering (Eng) | electrical engineering |
| Law | international law, jurisprudence, professional law |
| Health | anatomy, clinical knowledge, college medicine, human aging, nutrition, professional medicine, virology |

Table 13: MMLU subjects and tasks.

## H   CONSTRAINED BAYESIAN OPTIMIZATION

In this section, we demonstrate the corresponding results for all datasets. Figure 6 presents the relationship of Negative Log-Likelihood and Expected Calibration Error with optima choice explicitly marked.

We additionally present the ARC-Easy Pareto charts in Figure 7, using the same visualization scheme as WinoGrande-M (left: ACC vs. NLL; right: ACC vs. ECE).

Table 14 lists non-dominated candidates (w.r.t. ACC↑, NLL↓, ECE↓) returned by constrained BO for all datasets.

We select the final operating point by proximity to the empirical Pareto front (ties broken by lower NLL).

We tune learning rate $\eta$ and weight decay $\lambda$ on a bounded domain $\mathcal{X} \subset \mathbb{R}^2$:

$$\min_{\mathbf{x} \in \mathcal{X}} \mathbf{f}(\mathbf{x}) = \big(f_1(\mathbf{x}), f_2(\mathbf{x}), f_3(\mathbf{x})\big) = \big(\text{ECE, NLL, } -\text{ACC}\big), \tag{41}$$

with inequality constraints

$$c_j(\mathbf{x}) \leq 0, \quad j = 1, \ldots, J. \tag{42}$$

$$y_m(\mathbf{x}) = f_m(\mathbf{x}) + \varepsilon_m, \quad \varepsilon_m \sim \mathcal{N}(0, \sigma_m^2), \qquad \tilde{y}_j(\mathbf{x}) = c_j(\mathbf{x}) + \tilde{\varepsilon}_j, \ \tilde{\varepsilon}_j \sim \mathcal{N}(0, \tilde{\sigma}_j^2). \tag{43}$$

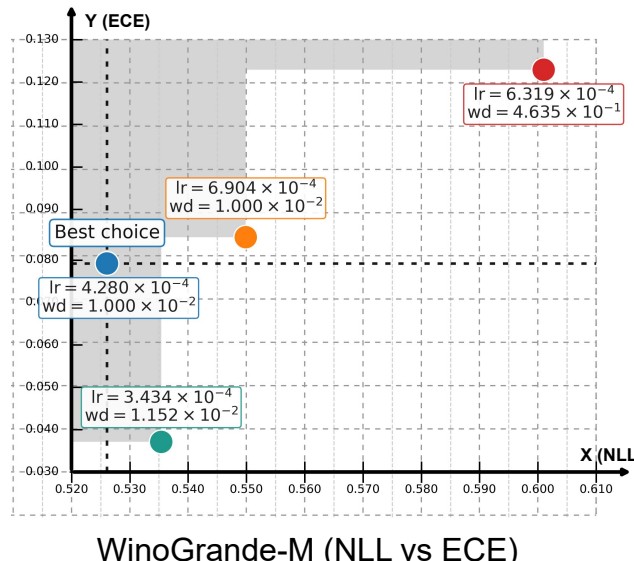

Figure 6: Pareto analysis on the WinoGrande-M dataset (NLL vs ECE). Each point denotes a hyperparameter pair $(\mathrm{lr}, \mathrm{wd})$. Gray regions show dominated solutions, and the cross marks the "Best choice" near the Pareto front.

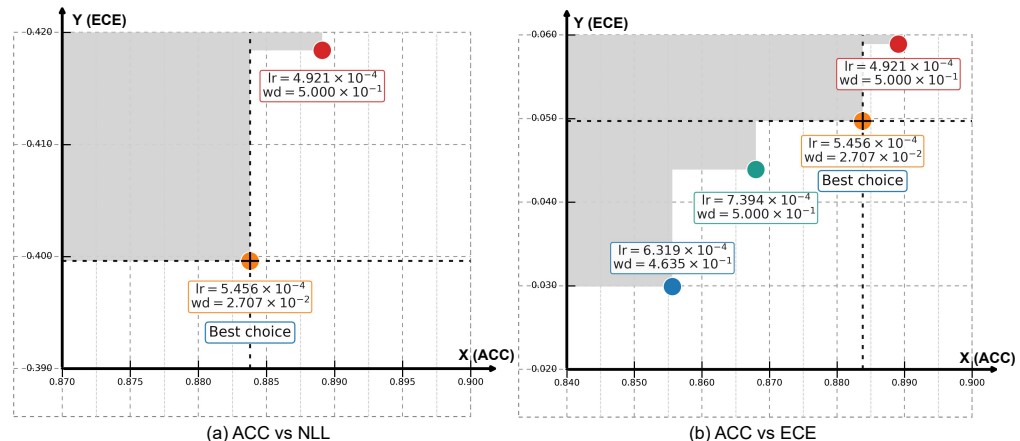

Figure 7: Pareto analysis on the ARC-Easy dataset (ACC vs NLL and ACC vs ECE). Each point denotes a hyperparameter pair $(\mathrm{lr}, \mathrm{wd})$. Gray regions show dominated solutions, and the cross marks the "Best choice" near the Pareto front.

For each objective/constraint:

$$f_m \sim \mathcal{GP}\big(\mu_m, k_m\big), \qquad c_j \sim \mathcal{GP}\big(\mu_j^{(c)}, k_j^{(c)}\big). \tag{44}$$

Given data $\mathcal{D}$ and a candidate batch $X = \{\mathbf{x}^{(q)}\}_{q=1}^{Q}$,

$$\mathbf{f}_m(X) \mid \mathcal{D} \sim \mathcal{N}\big(\boldsymbol{\mu}_{m|n}(X), \boldsymbol{\Sigma}_{m|n}(X)\big), \tag{45}$$

$$\boldsymbol{\mu}_{m|n}(X) = \mu_m(X) + K_m(X, X_{1:n})\big(K_m + \sigma_m^2 I\big)^{-1}\big(\mathbf{y}_m - \mu_m(X_{1:n})\big), \tag{46}$$

$$\boldsymbol{\Sigma}_{m|n}(X) = K_m(X, X) - K_m(X, X_{1:n})\big(K_m + \sigma_m^2 I\big)^{-1} K_m(X_{1:n}, X), \tag{47}$$

and analogously for constraints.

Table 14: Full Pareto candidates after constrained BO (all datasets combined).

| Dataset | Cand. | Acc | NLL | ECE (%) | LR | Weight Decay |
|---|---|---|---|---|---|---|
| WinoGrande-M | 1 | 0.7753 | 0.6010 | 12.29 | $6.319 \times 10^{-4}$ | $4.635 \times 10^{-1}$ |
| WinoGrande-M | 2 | 0.7682 | 0.5499 | 8.34 | $6.904 \times 10^{-4}$ | $1.000 \times 10^{-2}$ |
| WinoGrande-M | 3 | 0.7642 | 0.5329 | 8.15 | $6.319 \times 10^{-4}$ | $4.635 \times 10^{-1}$ |
| WinoGrande-M | 4 | 0.7634 | 0.5260 | 7.92 | $4.280 \times 10^{-4}$ | $1.000 \times 10^{-2}$ |
| WinoGrande-M | 5 | 0.7342 | 0.5354 | 3.71 | $3.434 \times 10^{-4}$ | $1.152 \times 10^{-2}$ |
| WinoGrande-S | 1 | 0.7445 | 0.6062 | 12.06 | $9.792 \times 10^{-5}$ | $9.339 \times 10^{-2}$ |
| WinoGrande-S | 2 | 0.7342 | 0.5870 | 9.90 | $7.832 \times 10^{-5}$ | $8.0912 \times 10^{-2}$ |
| WinoGrande-S | 3 | 0.7294 | 0.5474 | 2.74 | $9.792 \times 10^{-5}$ | $9.339 \times 10^{-2}$ |
| WinoGrande-S | 4 | 0.7033 | 0.5740 | 2.42 | $8.810 \times 10^{-5}$ | $7.783 \times 10^{-2}$ |
| ARC-C | 1 | 0.6959 | 0.8616 | 6.10 | $4.955 \times 10^{-5}$ | $2.056 \times 10^{-1}$ |
| ARC-C | 2 | 0.6858 | 0.8556 | 9.16 | $9.581 \times 10^{-5}$ | $5.000 \times 10^{-1}$ |
| ARC-C | 3 | 0.6014 | 1.0015 | 4.41 | $6.081 \times 10^{-5}$ | $3.141 \times 10^{-1}$ |
| ARC-E | 1 | 0.8891 | 0.4184 | 5.89 | $4.921 \times 10^{-4}$ | $5.000 \times 10^{-1}$ |
| ARC-E | 2 | 0.8838 | 0.3996 | 4.97 | $5.456 \times 10^{-4}$ | $2.707 \times 10^{-2}$ |
| ARC-E | 3 | 0.8680 | 0.4495 | 4.39 | $7.394 \times 10^{-4}$ | $5.000 \times 10^{-1}$ |
| ARC-E | 4 | 0.8556 | 0.4797 | 2.99 | $6.319 \times 10^{-4}$ | $4.635 \times 10^{-1}$ |
| OBQA | 1 | 0.8380 | 0.5499 | 7.18 | $1.180 \times 10^{-3}$ | $4.635 \times 10^{-1}$ |
| OBQA | 2 | 0.8280 | 0.5411 | 5.84 | $2.135 \times 10^{-4}$ | $2.025 \times 10^{-1}$ |
| BoolQ | 1 | 0.8641 | 0.2841 | 3.10 | $5.421 \times 10^{-4}$ | $3.582 \times 10^{-1}$ |
| BoolQ | 2 | 0.8572 | 0.3025 | 2.54 | $4.987 \times 10^{-4}$ | $2.041 \times 10^{-1}$ |
| BoolQ | 3 | 0.8490 | 0.3152 | 1.86 | $6.103 \times 10^{-4}$ | $5.000 \times 10^{-1}$ |

With independent constraints,

$$\mathrm{PoF}(X) \approx \prod_{q=1}^{Q} \prod_{j=1}^{J} \Phi \left( \frac{-\mu_{j|n}^{(c)}(\mathbf{x}^{(q)})}{\sqrt{\Sigma_{j|n}^{(c)}(\mathbf{x}^{(q)}, \mathbf{x}^{(q)})}} \right), \tag{48}$$

where $\Phi$ is the standard normal CDF.

**q-NEHVI acquisition (constrained).** Let $\mathbf{r}$ be the reference point and $\mathrm{HV}(\cdot; \mathbf{r})$ the dominated hypervolume. Define

$$\alpha_{\text{q-NEHVI}}(X) = \mathbb{E}\left[ \left( \mathrm{HV}(\tilde{\mathcal{P}} \cup \mathbf{F}(X); \mathbf{r}) - \mathrm{HV}(\tilde{\mathcal{P}}; \mathbf{r}) \right)_{+} \mathbb{I}\{\mathbf{C}(X) \leq \mathbf{0}\} \ \Big| \ \mathcal{D} \right], \tag{49}$$

where $\tilde{\mathcal{P}}$ is a sample of the posterior Pareto set (feasible and non-dominated), $\mathbf{F}(X)$ stacks the objective draws, and $\mathbf{C}(X)$ the constraint draws.

With Cholesky factors $\mathbf{L}_{m|n}(X)$ of equation 47,

$$\mathbf{F}_{m}^{(t)}(X) = \boldsymbol{\mu}_{m|n}(X) + \mathbf{L}_{m|n}(X)\mathbf{z}_{m}^{(t)}, \quad \mathbf{z}_{m}^{(t)} \sim \mathcal{N}(\mathbf{0}, I), \tag{50}$$

and similarly $\mathbf{C}^{(t)}(X)$. Sampling $\tilde{\mathcal{P}}^{(t)}$ from the posterior of historical designs, the MC estimator is

$$\widehat{\alpha}_{\text{q-NEHVI}}(X) = \frac{1}{T} \sum_{t=1}^{T} \left( \mathrm{HV}(\tilde{\mathcal{P}}^{(t)} \cup \mathbf{F}^{(t)}(X); \mathbf{r}) - \mathrm{HV}(\tilde{\mathcal{P}}^{(t)}; \mathbf{r}) \right)_{+} \mathbb{I}\{\mathbf{C}^{(t)}(X) \leq \mathbf{0}\}. \tag{51}$$

A PoF-weighted variant:

$$\widehat{\alpha}_{\text{q-NEHVI}}^{(\text{PoF})}(X) = \left( \frac{1}{T} \sum_{t=1}^{T} \left( \text{HV}(\tilde{\mathcal{P}}^{(t)} \cup \mathbf{F}^{(t)}(X); \mathbf{r}) - \text{HV}(\tilde{\mathcal{P}}^{(t)}; \mathbf{r}) \right)_+ \right) \cdot \text{PoF}(X). \tag{52}$$

And then we can get the reference point as follows:

$$\mathbf{r} = (r_1, r_2, r_3), \qquad r_m < \min_{\text{historical feasible}} f_m. \tag{53}$$

Table 15: Trainable parameters corresponding to different ranks.

| Rank | 4 | 9 | 16 | 32 | 64 | 128 |
|---|---|---|---|---|---|---|
| **Trainable Parameters** | 3.4M | 4.9M | 7.0M | 12M | 22M | 43M |

## I  USE OF LLMS

We used LLMs as general-purpose tools to assist in polishing the writing.

| Category | Hyperparameter (Value) |
|---|---|
| **Model & Tokenizer** | |
| Model name | `Qwen/Qwen3-30B-A3B-Instruct-2507` |
| Max input length | 1024 |
| BF16 precision | True |
| FP16 precision | False |
| Load in 8-bit | False |
| Gradient checkpointing | Optional (default: disabled) |
| Pad token | EOS token |
| **Training** | |
| Epochs | 1 |
| Batch size (per device) | 1 |
| Gradient accumulation steps | 1 |
| Learning rate | 1e–5 |
| Warmup steps | 100 |
| Optimizer | AdamW (PyTorch) |
| Max grad norm | 1.0 |
| Logging steps | 50 |
| Save strategy | Per epoch |
| Save total limit | 4 |
| Disable tqdm | False |
| **Dataset & Prompt** | |
| Dataset | `DigitalLearningGmbH/MATH-lighteval` |
| Split | train |
| Prompt type | qwen25-math-cot |
| Apply chat template | True |
| Max tokens per call | 1534 |
| **Deterministic LoRA** | |
| LoRA rank $r$ | 8 |
| LoRA $\alpha$ | 16 |
| LoRA dropout | 0.05 |
| Target modules | q_proj, v_proj |
| Bias | none |
| Task type | Causal LM |
| **Bayesian LoRA (ours)** | |
| Bayesian LoRA layer | `BayesianLoRALayer` |
| Bayesian rank ($r_{\text{bayes}}$) | 9 |
| Linear layer override | `nn.Linear` $\rightarrow$ `BayesianLoRALayer` |
| Posterior treatment | Variational / structured low-rank |
| Uncertainty sampling | Enabled (Bayesian forward samples) |
| Merge for inference | supported via `merge_and_unload` |
| **Evaluation (unused in this script)** | |
| Temperature | 0 |
| Top-p | 1.0 |
| Num sampling | 1 |
| vLLM eval | Optional |
| Eval batch size | 1 |

Table 16: Training hyperparameters for deterministic LoRA and Bayesian LoRA fine-tuning on the MATH dataset. Bayesian LoRA uses a structured Bayesian low-rank update with rank $r_{\text{bayes}} = 9$.

