# OpenReview forum: "Bayesian-LoRA: Gaussian Process Modeling for Large Language Models"
_ICLR.cc/2026/Conference — Submitted to ICLR 2026_

### Official Review · Reviewer_Sx7d · 2025-10-28

**Soundness:** 4
**Presentation:** 3
**Contribution:** 3
**Rating:** 4
**Confidence:** 3

**Summary:**

The paper proposes Bayesian-LoRA, integrating Sparse Gaussian Process (SGP) into LoRA with normalizing flow (adding ~0.42M params) to stabilize training . It improves LLM calibration (reduces ECE/NLL) post-fine-tuning (avoids overconfidence) while retaining LoRA’s efficiency . Experiments on LLaMA 2-7B across 6 commonsense benchmarks and WikiText-2 show it outperforms baselines in accuracy/calibration.

**Strengths:**

1.The paper is well organized and well written.

2.The authors present a well-motivated approach.

3.It conducts numerous experiments, validates the experimental results on models of various series and sizes, and covers a wide range of evaluation tasks

**Weaknesses:**

1. In line 53, the abbreviation “MAP” should be explained when it first appears, for instance as *Maximum A Posteriori*, to help readers better understand the term.

2. The paper lacks experiments on larger-scale models.
   Evaluating the proposed method on models of different or larger scales would help demonstrate its robustness and practical applicability.

3. According to Table 3, the proposed method incurs approximately 1.2× higher training cost and 1.5–2.7× higher inference cost compared to LoRA. When compared with deterministic PEFT methods such as AdaLoRA[1], DoRA[2], and PiSSA[3], the proposed approach does not seem to provide clear advantages. Since the paper does not include direct comparisons with these methods, it remains unclear whether the proposed approach has distinct limitations or strengths. The authors are encouraged to include more detailed comparisons and analysis to clarify this point.

[1] AdaLoRA: Adaptive Budget Allocation for Parameter-Efficient Fine-Tuning

[2] DoRA: Weight-Decomposed Low-Rank Adaptation

[3] PiSSA: Principal Singular Values and Singular Vectors Adaptation of Large Language Models

**Questions:**

see weaknesses

---

> ### Author Response · Authors · 2025-11-25
> **Responses**
>
> **Q1**: In line 53, the abbreviation “MAP” should be explained when it first appears, for instance as Maximum A Posteriori, to help readers better understand the term.
>
> **A1**:  We thank the reviewer for pointing this out. We been corrected in the revision(Page 1, line 53);  We explicitly explain that “MAP” refers to the deterministic LoRA, and we will avoid using this label elsewhere to prevent ambiguity
>
> ---
>
> **Q2**: The paper lacks experiments on larger-scale models. Evaluating the proposed method on models of different or larger scales would help demonstrate its robustness and practical applicability.
>
> **A2**:  We appreciate the reviewers’ feedback.  We have now conducted additional evaluations on substantially larger architectures, including Qwen2.5-14B-Instruct and Qwen3-30B-A3B-Instruct-2507, using the MATH reasoning benchmark.
>
>
> #### Table: Performance comparison on the **MATH** dataset using large-scale models  We report **CoT Negative Log-Likelihood (NLL)**, **CoT Expected Calibration Error (ECE)**, and **final answer accuracy**. Baseline results come from standard fine-tuning, and our Bayesian-LoRA approach. The hyperparameters for training configuration are reported in Table15.
>
> | **Model (Zero-shot)**            | **Method**   | **CoT-NLL ↓** | **CoT-ECE ↓** | **Answer Acc. ↑** |
> |----------------------------------|--------------|---------------|----------------|--------------------|
> | **Qwen2.5-14B-Instruct**         | Baseline FT  | 2.165         | 12.2           | 49.8               |
> |                                  | **Bayesian-Lora**  | **0.513**     | **5.81**       | **51.1**           |
> | **Qwen3-30B-A3B-Instruct-2507**  | Baseline FT  | 1.096         | 8.96           | 61.8               |
> |                                  | **Bayesian-Lora**  | **0.721**     | **6.32**       | **61.9**           |
>
>
>
>
> Our implementation of Bayesian-LoRA is currently being merged into PEFT as a new variant. And then you can use it directly by:
> ```
> pip install peft
> ```
> and enabled simply by registering our Bayesian-LoRA layer:
> ```
> lora_config = LoraConfig(
>     r=args.lora_r,
>     lora_alpha=args.lora_alpha,
>     target_modules=target_modules,
>     lora_dropout=args.lora_dropout,
>     bias="none",
>     task_type="CAUSAL_LM",
> )
>
> # add the following line for Bayesian-LoRA
> lora_config._register_custom_module({
>     nn.Linear: BayesianLoRALayer
> })
>
> model = get_peft_model(model, lora_config)
> ```
> ---
>
> **Q3**: According to Table 3, the proposed method incurs approximately 1.2× higher training cost and 1.5–2.7× higher inference cost compared to LoRA. When compared with deterministic PEFT methods such as AdaLoRA[1], DoRA[2], and PiSSA[3], the proposed approach does not seem to provide clear advantages. Since the paper does not include direct comparisons with these methods, it remains unclear whether the proposed approach has distinct limitations or strengths. The authors are encouraged to include more detailed comparisons and analysis to clarify this point.
>
>
> **A3**:   We thank the reviewer for the insightful comment. In the Bayesian, the extra computational cost is fundamentally unavoidable. There is no “free lunch’’ when estimating predictive uncertainty. Our goal in this work is precisely to strike a favorable trade-off between **uncertainty quality** and **computational efficiency**. The proposed SGP-based Bayesian LoRA significantly reduces the overhead typically associated with Bayesian neural networks, and providing substantially better uncertainty estimates compared to existing Bayesian or stochastic PEFT methods (as shown in Table 3–4).
> Deterministic PEFT approaches such as AdaLoRA, DoRA, and PiSSA are designed for accuracy and efficiency, but they do **not** provide meaningful uncertainty estimates, making a direct comparison less informative. Nonetheless, our results demonstrate that Bayesian LoRA offers a compelling balance: modest overhead relative to LoRA, while yielding strong gains in calibration and risk-aware performance. We have added clarifications in the revision to emphasize this distinction and the strengths of the proposed approach.

---

### Official Review · Reviewer_3vY8 · 2025-10-28

**Soundness:** 2
**Presentation:** 3
**Contribution:** 2
**Rating:** 6
**Confidence:** 4

**Summary:**

This paper addresses the problem of calibration deterioration in LLMs following PEFT, such as LoRA, especially on small datasets. The authors propose Bayesian-LoRA, a method that applies a Sparse Gaussian Process (SGP) to the LoRA update mechanism. This approach models the distribution over the low-rank update weights using a small set of inducing variables. To improve the flexibility of the variational posterior, the SGP is integrated with a normalizing flow. Experiments fine-tuning LLaMA 2-7B on several benchmarks show that Bayesian-LoRA improves calibration metrics without sacrificing accuracy. The method maintains parameter efficiency, adding only ~0.42M parameters, and incurs modest computational overhead compared to standard LoRA.

**Strengths:**

S1: Fine-tuned LLMs often become miscalibrated and overconfident, especially in safety-critical use cases. Addressing this behavior has high practical relevance.
S2: Introducing Bayesian modeling into the LoRA subspace is conceptually appealing and improves scalability, addressing limitations of parameter-space Bayesian methods.
S3: Normalizing Flow enriches the posterior family. This avoids overly restrictive Gaussian assumptions and allows the posterior to capture multi-modal or heavy-tailed structures — a thoughtful design choice.

**Weaknesses:**

W1: Method complexity and stacking of components. The proposed approach combines LoRA + SGP + Normalizing Flow. While innovative, the contributions of each component are not fully disentangled. More fine-grained ablations are needed.
W2: The experimental evaluation is limited in scale, primarily using mid-sized models (e.g., 7B). Additional results on larger models (e.g., 13B or 70B) would strengthen the paper’s claims regarding scalability and general applicability to LLM fine-tuning.
W3: The connection between the "Variational Sparse Inducing Weight Model" preliminaries (Sec. 2.2) and the final Bayesian-LoRA method (Sec. 3.1) is confusing. Section 2.2 defines the SGP as modeling the conditional mean $M_W(U)$ of the full weight matrix $W$ (Eq. (4), (6)). However, Section 3.1 and Figure 1 imply the SGP is used to model the LoRA matrices $A$ and $B$, which compose the update $\Delta W$. This is a critical disconnect. Also, algorithm 1 deepens this confusion. It suggests that both $\bar{A}l$ and $\bar{B}l$ are generated from the same inducing variable sample $U^{(s)}$ using different projection matrices ($T^A$, $T^B$). This implementation detail is not derived or explained in the main text, making it unclear how $U$ is shared and why this is the correct formulation.
W4: The rank $r$ of the baseline LoRA (MAP) model (listed at 4.48M parameters in Table 3) is not specified.
W5: The paper needs to explicitly state: is the SGP applied to $A$ and $B$ separately, or jointly from one $U$? The current presentation, mixing preliminaries on $W$ (Sec 2.2) with an implementation on $A$ and $B$ (Alg. 1), is difficult to follow.

**Questions:**

Please refer to Weakness.

**Details Of Ethics Concerns:**

N.A.

---

> ### Author Response · Authors · 2025-11-25
> **Responses**
>
> **Q1**: While innovative, the contributions of each component are not fully disentangled; More fine-grained ablations are needed.
>
>
> **A1**: We thank the review raising this problem.  We have conducted some extended hyperparameter study covering LoRA rank (Figure 2, Section 4.4), learning rate(Figure 3, Table 3 and Table 14); We will include these interactions between  and other hyperparameters as detail as possible in the paper. And our method is not a simple stacking; it works because LoRA’s low-rank form naturally matches the SGP’s matrix-normal/Kronecker structure.
> LoRA parameterizes the update as
>
> \begin{equation}
> \Delta W = B A .
> \end{equation}
>
> In Sec. 2.2, the sparse-GP model defines the conditional mean of a weight matrix by
>
> \begin{equation}
> M_W(U) = T_r\ U\ T_c, where \ \mathbf{T_r} = Z_r^\top K_r^{-1} \; \mathbf{T_c} = K_c^{-1} Z_c
> \end{equation}
>
> where the projection operators $T_r$ and $T_c$are derived from the inducing
> row/column covariances; Z is covariances matrix.  And it can be treated as $B = T_r\ U$, $A = T_c$ that exactly matches the LoRa structure.
>
>
> **Q2**: The experimental evaluation is limited in scale, primarily using mid-sized models (e.g., 7B). Additional results on larger models (e.g., 13B or 70B) would strengthen the paper’s claims regarding scalability and general applicability to LLM fine-tuning.
>
> **A2**: We appreciate the reviewers’ feedback. Actually, all reviewers pointed this out,  including Reviewer SG6J Q5, Reviewer pPT1 Q1 and Reviewer Sx7d Q2. In response, we have now conducted additional evaluations on substantially larger architectures, including Qwen2.5-14B-Instruct and Qwen3-30B-A3B-Instruct-2507, using the MATH reasoning benchmark.
>
> #### Table: Performance comparison on the **MATH** dataset using large-scale models  We report **CoT Negative Log-Likelihood (NLL)**, **CoT Expected Calibration Error (ECE)**, and **final answer accuracy**. Baseline results come from standard fine-tuning, and our Bayesian-LoRA approach. The hyperparameters for training configuration are reported in Table15.
>
> | **Model (Zero-shot)**            | **Method**   | **CoT-NLL ↓** | **CoT-ECE ↓** | **Answer Acc. ↑** |
> |----------------------------------|--------------|---------------|----------------|--------------------|
> | **Qwen2.5-14B-Instruct**         | Baseline FT  | 2.165         | 12.2           | 49.8               |
> |                                  | **Bayesian-Lora**  | **0.513**     | **5.81**       | **51.1**           |
> | **Qwen3-30B-A3B-Instruct-2507**  | Baseline FT  | 1.096         | 8.96           | 61.8               |
> |                                  | **Bayesian-Lora**  | **0.721**     | **6.32**       | **61.9**           |
>
>
>
>
> Our implementation of Bayesian-LoRA is currently being merged into PEFT as a new variant. And then you can use it directly by:
> ```
> pip install peft
> ```
> and enabled simply by registering our Bayesian-LoRA layer:
> ```
> lora_config = LoraConfig(
>     r=args.lora_r,
>     lora_alpha=args.lora_alpha,
>     target_modules=target_modules,
>     lora_dropout=args.lora_dropout,
>     bias="none",
>     task_type="CAUSAL_LM",
> )
>
> # add the following line for Bayesian-LoRA
> lora_config._register_custom_module({
>     nn.Linear: BayesianLoRALayer
> })
>
> model = get_peft_model(model, lora_config)
> ```
>
> **Q3**: The connection between the “Variational Sparse Inducing Weight Model” preliminaries (Sec. 2.2) and the final Bayesian-LoRA method (Sec. 3.1) is confusing. Section 2.2 defines the SGP as modeling the conditional mean ( M_W(U) ) of the full weight matrix ( W ) (Eq. (4), (6)). However, Section 3.1 and Figure 1 imply the SGP is used to model the LoRA matrices ( A ) and ( B ), which compose the update ( \Delta W ). Algorithm 1 further deepens the confusion: it suggests that both ( \tilde{A} ) and ( \tilde{B} ) are generated from the same inducing variable sample ( U^{(s)} ) using different projection matrices ( T^A, T^B ).
>
>
> **A3**: Thank you for pointing this out. In our method, the sparse inducing weight model from Sec. 2.2 is *only* applied to the **LoRA update**.   And our method is not a simple stacking; it works  because LoRA’s low-rank form naturally matches the SGP’s matrix-normal/Kronecker structure.
>
> LoRA parameterizes the update as
>
> \begin{equation}
> \Delta W = B A .
> \end{equation}
>
> In Sec. 2.2, the sparse-GP model defines the conditional mean of a weight matrix by
>
> \begin{equation}
> M_W(U) = T_r\ U\ T_c, where \; \mathbf{T_r} = Z_r^\top K_r^{-1} \; \mathbf{T_c} = K_c^{-1} Z_c
> \end{equation}
>
> where the projection operators $T_r$ and $T_c$are derived from the inducing
> row/column covariances; Z is covariances matrix.  And it can be treated as $B = T_r\ U$, $A = T_c$ .
>
> This is exactly a Kronecker-structured low-rank factorization that maps the inducing matrix $U$ into the weight space.
> $ \tilde{A} $and $ \tilde{B} $ are from the same U as it represents the left and right covariance of U(Kronecker structure) and we will clarify it in the paper.

---

> ### Author Response · Authors · 2025-11-25
> **responses part 2**
>
> **Q4**: The rank  of the baseline LoRA (MAP) model (listed at 4.48M parameters in Table 3) is not specified.
>
> **A4**: We thank the reviewer for pointing this out. This was our oversight ,  the rank of the baseline LoRA (MAP) model is also 9, matching our Bayesian LoRA setup. We have updated the paper to explicitly include this information. (page 6, line 277)
>
>
> **Q5**: The paper needs to explicitly state: is the SGP applied to  and  separately, or jointly from one ? The current presentation, mixing preliminaries on  (Sec 2.2) with an implementation on  and  (Alg. 1), is difficult to follow.
>
> **A5**: We thank the reviewer for the helpful comment. The SGP is jointly applied through a single latent matrix $U$, rather than separately on $A$ and $B$. Specifically, the inducing-row and inducing-column projections $(T_r, T_c)$ act on a shared $U$, providing the LoRA-style update
>
> \begin{equation}
> M_W(U) = T_r\, U\, T_c
> \end{equation}
>
> We have clarified this in Sec. 2.2 and Alg. 1 to avoid mixing preliminaries with implementation details.

---

### Official Review · Reviewer_pPT1 · 2025-10-31

**Soundness:** 4
**Presentation:** 3
**Contribution:** 2
**Rating:** 4
**Confidence:** 2

**Summary:**

The paper proposes Bayesian-LoRA: instead of a deterministic LoRA update, it introduces a SGP over an inducing variable (U) defined in the low-rank LoRA subspace. A Kronecker structure along row/column dimensions maps (U) to the means of (A,B).

**Strengths:**

- Probabilizing LoRA’s low-rank update via a Kronecker SGP and correcting with a small flow is clean. Compared to parameter-space Laplace/VI (LA/LLLA, BLoB), this leans toward a more “function-space-like” view, with a simple closed-form KL and a straightforward training recipe.

**Weaknesses:**

To be honest, I am not familiar with the core knowledge involved in the paper. Perhaps for professionals in the field, this is an excellent paper, so I am willing to refer to other reviewers' opinions to adjust my score. The following are just some personal suggestions of mine.

- Add at least one modern backbone (e.g., Llama-3-8B-Instruct) and 1–2 broader benchmarks (HellaSwag, GSM8K, or full MMLU). Report ACC/NLL/ECE and latency vs. number of samples?

- Although the paper aims to move away from parameter-space approximations, the method still samples weight updates (albeit structured) rather than directly ensuring parameterization-invariant function perturbations. The “function-space advantage” needs either stronger theory (e.g., demonstrating invariance benefits under reparameterization) or targeted experiments (e.g., holding a function-distance budget constant across methods)?

**Questions:**

see weekness above

---

> ### Author Response · Authors · 2025-11-25
> **Responses**
>
> **Q1**: Add at least one modern backbone (e.g., Llama-3-8B-Instruct) and 1–2 broader benchmarks (HellaSwag, GSM8K, or full MMLU). Report ACC/NLL/ECE and latency vs. number of samples?
>
> **A1**: Thank you for pointing this out. We have now conducted more comprehensive experiments on larger architectures, including Qwen2.5-14B-Instruct and Qwen3-30B-A3B-Instruct-2507, on the MATH reasoning benchmark.
> #### Table: Performance comparison on the **MATH** dataset using large-scale models  We report **CoT Negative Log-Likelihood (NLL)**, **CoT Expected Calibration Error (ECE)**, and **final answer accuracy**. Baseline results come from standard fine-tuning, and our Bayesian-LoRA approach. The hyperparameters for training configuration are reported in Table15.
>
> | **Model (Zero-shot)**            | **Method**   | **CoT-NLL ↓** | **CoT-ECE ↓** | **Answer Acc. ↑** |
> |----------------------------------|--------------|---------------|----------------|--------------------|
> | **Qwen2.5-14B-Instruct**         | Baseline FT  | 2.165         | 12.2           | 49.8               |
> |                                  | **Bayesian-Lora**  | **0.513**     | **5.81**       | **51.1**           |
> | **Qwen3-30B-A3B-Instruct-2507**  | Baseline FT  | 1.096         | 8.96           | 61.8               |
> |                                  | **Bayesian-Lora**  | **0.721**     | **6.32**       | **61.9**           |
>
>
>
>
> Our implementation of Bayesian-LoRA is currently being merged into PEFT as a new variant. And then you can use it directly by:
> ```
> pip install peft
> ```
> and enabled simply by registering our Bayesian-LoRA layer:
> ```
> lora_config = LoraConfig(
>     r=args.lora_r,
>     lora_alpha=args.lora_alpha,
>     target_modules=target_modules,
>     lora_dropout=args.lora_dropout,
>     bias="none",
>     task_type="CAUSAL_LM",
> )
>
> # add the following line for Bayesian-LoRA
> lora_config._register_custom_module({
>     nn.Linear: BayesianLoRALayer
> })
>
> model = get_peft_model(model, lora_config)
> ```
>
> ---
>
>
> **Q2**: Although the paper aims to move away from parameter-space approximations, the method still samples weight updates (albeit structured) rather than directly ensuring parameterization-invariant function perturbations. The “function-space advantage” needs either stronger theory (e.g., demonstrating invariance benefits under reparameterization) or targeted experiments (e.g., holding a function-distance budget constant across methods)?
>
> **A2** :Thank you for raising this point. we place a sparse GP prior on inducing variables and derive the distribution over $\Delta W$ by the conditional mean $ M_W(U)$ and the flow, so the apparent “weight sampling” is a reparameterization of a GP defined in the LoRA subspace, with KL invariance under  $ T_\phi$  formalized in Proposition 3.1. We have softened the wording around the “function-space advantage” and added a brief discussion that fully parameterization-invariant function perturbations and function-distance–controlled comparisons are promising directions for future work.

---

> > ### Comment · Reviewer_pPT1 · 2025-11-25
> >
> > Thank you for the reply. The author's response has addressed my concerns, and I will increase my score. Good luck.

---

> > > ### Author Response · Authors · 2025-11-25
> > >
> > > Thank you very much for the positive update. We appreciate your support!

---

### Official Review · Reviewer_SG6J · 2025-11-01

**Soundness:** 2
**Presentation:** 2
**Contribution:** 2
**Rating:** 2
**Confidence:** 3

**Summary:**

This paper focuses on addressing LLM overconfident predictions, especially when LLMs are fine-tuned on small datasets or experience domain shifts. The authors propose Bayesian-LoRA, combining a sparse Gaussian Process (SGP) and normalizing flow. They fine-tune LLaMA-2-7B on commonsense reasoning tasks (e.g., WinoGrande-S, ARC-C/E, OBQA, BoolQ) and generative tasks (WikiText-2), reporting improvements in calibration metrics without sacrificing accuracy and without introducing significant additional cost.

**Strengths:**

1. The paper focus on uncertainty and calibration, which is important for safety-critical domain.
2. The method is concepttally novel. combining a sparse Gaussian Process (SGP) and normalizing flow.
3. The added training time and memory overhead are modest (training time ~1.2×, adding only ≈ 0.42M parameters)
4. They reporting improvements in calibration metrics without sacrificing accuracy and without introducing significant additional cost. They also test out-of-distribution generalization (i.e., robustness when input distributions shift)

**Weaknesses:**

**1. Weakness in Method:**
  - The reported improvements are marginal. For example, in Table 5, the reported ECE outperforms baselines on only 3 out of 6 OOD tasks and does not perform well on ID tasks.
  - The paper recommends $S = 2-4$ to avoid cost, but does not quantify how predictive quality (ACC/NLL/ECE) degrades for small $S$ or how latency/throughput scale for larger $S$. Only a qualitative statement (`"slight improvement with higher $S$") is given. This makes the accuracy–latency trade-off hard to judge in practice. It also calls into question the effectiveness of the method, in general, more samples should yield more accurate uncertainty estimation.
  - Deeper flows ($L > 1$) give limited gains at extra cost, and the paper defaults to $L = 1$. This raises the question of when (if ever) the flow is truly beneficial compared to the base SGP.
  - Can this method be applied to full fine-tuning? Or to broader adapter families?

**2. Weakness in Evaluation:**
  - Evaluation is limited to six small MCQ benchmarks (WinoGrande-S/M, ARC-C/E, OBQA, BoolQ) and WikiText-2 for language modeling. More practical and challenging tasks, e.g., instruction-following, math reasoning, and coding which also require calibration, are missing.
  - Experiments are restricted to LLaMA-2-7B, more models should be included.
  - Only ECE and NLL are reported, offering little insight. Additional metrics / case studies / visualization should be provided.
  - The impact of different hyperparameters (e.g., LoRA rank, learning rate, training steps) on performance and efficiency is not well studied. Calibration and accuracy may depend strongly on the low-rank dimension $r$, and on the interaction between $r$ and other hyperparameters.

**3. Weakness in Writing:**
  - The writing in the paper is confusing. The paper labels standard LoRA as "MAP" without explicit reason.
  - Background on calibration is too brief for non-specialists. Readers unfamiliar with calibration may struggle to interpret the metrics and their practical meaning.
  - The specific role of each component, the rationale for choosing them, and how they interact are unclear. The method reads like two techniques simply stacked together.

**Questions:**

See Weakness

---

> ### Author Response · Authors · 2025-11-25
> **Part 1**
>
> **Q1**: The reported improvements are marginal. For example, in Table 5, the reported ECE outperforms baselines on only 3 out of 6 OOD tasks and does not perform well on ID tasks.
>
> **A1**: We thank the reviewer for the careful reading and constructive feedback and we are really sorry for not achieving SOTA for all datasets.  Our goal in this work is not to optimize for per-dataset SOTA on every metric, but rather to contribute a principled Bayesian framework for uncertainty estimation in LLMs. We acknowledge that on certain datasets/metrics, like ECE on some OOD datasets, we are not the best; however, across most settings we remain competitive in accuracy, achieve stronger accuracy and calibration.
>
> ---
>
> **Q2**:The paper recommends S=2-4 to avoid cost, but does not quantify how predictive quality (ACC/NLL/ECE) degrades for small or how latency/throughput scale for larger . Only a qualitative statement (`"slight improvement with higher ") is given. This makes the accuracy–latency trade-off hard to judge in practice. It also calls into question the effectiveness of the method, in general, more samples should yield more accurate uncertainty estimation.
>
> **A2**:
> Thank you for this comment.  We have now added a quantitative study of the trade-off between the number of MC samples $S$, predictive quality, and latency. When we increase $S$ from 1 to 10 on the in-distribution (ID) ARC dataset, inference time grows almost linearly, but ACC/NLL/ECE change only very slightly once $S \ge 2$, so additional samples bring negligible benefit compared to the cost. On the out-of-distribution (OOD) OBQA dataset, larger $S$ gives slightly improved and more stable NLL/ECE, but the gains beyond $S=4$ remain small relative to the 2–3× (or more) increase in latency. This is why we recommend $S=2$–$4$ as a practical setting that captures most of the benefit in uncertainty estimation and OOD detection while keeping computational overhead reasonable. **We have also added a subsection in Appendix C to report these results, including Table.7 , Table 8 and Figure 5**.
> By the way, the base model is fixed, and the sampling process occurs within the LoRA, which contains only 4 million parameters.
> At the same time, based on the vLLM concepts, the sampling here is actually a batch repetition. For example, sampling 4 times means overlaying 4 identical prompts, so overall it's just a single forward pass
>
>
> ### OOD (EP=2, STEP=843, train_mode=True)
>
> | S (MC samples) |   NLL   |  ACC   |  ECE   | avg_batch_time (s) |
> |----------------|---------|--------|--------|--------------------|
> | 1              | 1.0756  | 0.6333 | 0.1555 |  2.3997            |
> | 2              | 1.0723  | 0.6354 | 0.1476 |  4.7858            |
> | 3              | 1.0716  | 0.6333 | 0.1516 |  7.1729            |
> | 4              | 1.0707  | 0.6354 | 0.1480 |  9.5596            |
> | 5              | 1.0708  | 0.6313 | 0.1521 | 11.9460            |
> | 6              | 1.0704  | 0.6354 | 0.1488 | 14.3300            |
> | 7              | 1.0699  | 0.6333 | 0.1503 | 16.7140            |
> | 8              | 1.0697  | 0.6354 | 0.1482 | 19.0961            |
> | 9              | 1.0697  | 0.6354 | 0.1509 | 21.4805            |
> | 10             | 1.0702  | 0.6354 | 0.1491 | 23.8661            |
>
>
>
>
> ### ID (EP=2, STEP=843, train_mode=True)
>
> | S (MC samples) |   NLL   |  ACC   |  ECE   | avg_batch_time (s) |
> |----------------|---------|--------|--------|--------------------|
> | 1              | 0.4245  | 0.8697 | 0.0604 | 0.7610             |
> | 2              | 0.4245  | 0.8662 | 0.0631 | 1.5253             |
> | 3              | 0.4253  | 0.8662 | 0.0644 | 2.2888             |
> | 4              | 0.4252  | 0.8662 | 0.0605 | 3.0507             |
> | 5              | 0.4252  | 0.8680 | 0.0584 | 3.8133             |
> | 6              | 0.4248  | 0.8680 | 0.0572 | 4.5761             |
> | 7              | 0.4246  | 0.8662 | 0.0606 | 5.3384             |
> | 8              | 0.4246  | 0.8680 | 0.0588 | 6.1010             |
> | 9              | 0.4249  | 0.8680 | 0.0572 | 6.8632             |
> | 10             | 0.4246  | 0.8680 | 0.0599 | 7.6256             |

---

> ### Author Response · Authors · 2025-11-25
> **part 2**
>
> **Q3**: Deeper flows ($L > 1$) give limited gains at extra cost, and the paper defaults to $L = 1$. This raises the question of when (if ever) the flow is truly beneficial compared to the base SGP.
>
> **A3**: Thank you for this comment. We already include an ablation on the flow depth \(L\) in Table 4. The results show that deeper flows (\(L > 1\)) can have slightly better performance, but the improvements are relatively modest compared to the additional computational and memory cost. In contrast, the single-layer flow (\(L=1\) offers a favorable trade-off between performance gains and efficiency, and we therefore adopt it as our default setting.
>
> #### Table : **Ablation on flow depth L in the posterior transform \(T_\phi\) on OBQA Values are macro-averages (mean ± std over 3 seeds). Δ = current – value at L=1. ACC/ECE in percentage points; NLL is absolute difference. Efficiency is relative to standard LoRA (MAP)**
>
> | **Flow depth \(L\)** | **ACC ↑** (value) | **Δ vs. L=1** | **ECE ↓** (value) | **Δ vs. L=1** | **NLL ↓** (value) | **Δ vs. L=1** | **Train time (×MAP)** | **Peak mem. (×MAP)** |
> |----------------------|-------------------|---------------|--------------------|---------------|---------------------|----------------|-------------------------|------------------------|
> | **0 (pure SGP)**     | 79.0 ± 0.21       | -2.6          | 5.8 ± 0.13         | +0.1          | 0.58 ± 0.08         | +0.01          | 1.19                    | 1.002                  |
> | **1**                | 81.6 ± 0.10       | 0.0           | 5.7 ± 0.20         | 0.0           | 0.57 ± 0.12         | 0.00           | 1.23                    | 1.003                  |
> | **2**                | 80.8 ± 0.14       | -0.8          | 5.6 ± 0.09         | -0.1          | 0.52 ± 0.06         | -0.05          | 1.30                    | 1.008                  |
> | **4**                | 80.9 ± 0.08       | -0.7          | 4.9 ± 0.03         | -0.8          | 0.48 ± 0.13         | -0.09          | 1.38                    | 1.010                  |
>
> ---
>
>
>
> **Q4**: Can this method be applied to full fine-tuning? Or to broader adapter families?
>
> **A4**:
> Yes. It can be applied to full fine-tuning. We experimented with full fine-tuning by ResNet-18 in ImageNet, and it not only slightly improves accuracy, but also achieves better calibration and more reliable uncertainty estimates.
>
> ### Full Fine-Tuning on ResNet-18
>
> | Method | Setting | Accuracy (%) ↑ | NLL ↓ | ECE (%) ↓ | FLOPs | #Params (M) ↓ |
> |--------|---------|----------------|-------|------------|--------|----------------|
> | Deterministic⁺ | Full fine-tuning (baseline) | 69.68 | 1.12 | 5.25 | **3.62G** | 11.6M |
> | Ours (Matheron, 4 layers) | Partial-layer Bayesian fine-tuning | 70.17 | 1.13 | 6.86 | 8.09G | 8.48M |
> | **Ours (Matheron, all layers)** | **Full fine-tuning (all layers)** | **70.19** | **1.06** | **4.81** | 14.7G | **5.62M** |
>
> In principle, our method can be applied both to full fine-tuning and to broader adapter families but needs more resources (GPUs).

---

> ### Author Response · Authors · 2025-11-25
> **part 3**
>
> **Q5**: Evaluation is limited to six small MCQ benchmarks (WinoGrande-S/M, ARC-C/E, OBQA, BoolQ) and WikiText-2 for language modeling. More practical and challenging tasks, e.g., instruction-following, math reasoning, and coding which also require calibration, are missing.
>
> **A5**: Thank you for pointing this out. We have now conducted more comprehensive experiments on larger architectures, including Qwen2.5-14B-Instruct and Qwen3-30B-A3B-Instruct-2507, on the MATH reasoning benchmark.
> Our implementation of Bayesian-LoRA is currently being merged into PEFT as a new variant. And then you can use it directly by:
> ```
> pip install peft
> ```
> and enabled simply by registering our Bayesian-LoRA layer:
> ```
> lora_config = LoraConfig(
>     r=args.lora_r,
>     lora_alpha=args.lora_alpha,
>     target_modules=target_modules,
>     lora_dropout=args.lora_dropout,
>     bias="none",
>     task_type="CAUSAL_LM",
> )
>
> # add the following line for Bayesian-LoRA
> lora_config._register_custom_module({
>     nn.Linear: BayesianLoRALayer
> })
>
> model = get_peft_model(model, lora_config)
> ```
> We have also completed experiments on the large-scale models mentioned above.
>  The results on the MATH dataset are summarized below:
> #### Table: Performance comparison on the **MATH** dataset using large-scale models  We report **CoT Negative Log-Likelihood (NLL)**, **CoT Expected Calibration Error (ECE)**, and **final answer accuracy**. Baseline results come from standard fine-tuning, and our Bayesian-LoRA approach. The hyperparameters for training configuration are reported in Table15.
>
> | **Model (Zero-shot)**            | **Method**   | **CoT-NLL ↓** | **CoT-ECE ↓** | **Answer Acc. ↑** |
> |----------------------------------|--------------|---------------|----------------|--------------------|
> | **Qwen2.5-14B-Instruct**         | Baseline FT  | 2.165         | 12.2           | 49.8               |
> |                                  | **Bayesian-Lora**  | **0.513**     | **5.81**       | **51.1**           |
> | **Qwen3-30B-A3B-Instruct-2507**  | Baseline FT  | 1.096         | 8.96           | 61.8               |
> |                                  | **Bayesian-Lora**  | **0.721**     | **6.32**       | **61.9**           |
>
>
>
> ---
>
> **Q6**: Experiments are restricted to LLaMA-2-7B, more models should be included.
>
> **A6**: Thank you for raising this point. We fully agree that a broader set of architectures is necessary as all reviewers pointed this problem out. and as detailed in Q5, we have already extended our experiments to larger and more recent architectures, addressing this concern
>
> ---
> **Q7**: Only ECE and NLL are reported, offering little insight. Additional metrics / case studies / visualization should be provided.
>
> **A7**: We thank the reviewer for raising this point. In response, we have added CoT-NLL and CoT-ECE as chain-of-thought–level uncertainty metrics that provide deeper insight into the reliability and calibration of the reasoning process
>
> ---
>
>
> **Q8**:The impact of different hyperparameters (e.g., LoRA rank, learning rate, training steps) on performance and efficiency is not well studied. Calibration and accuracy may depend strongly on the low-rank dimension , and on the interaction between  and other hyperparameters.
>
> **A8**: We thank the reviewer for raising this important point. We have conducted some extended hyperparameter study covering LoRA rank (Figure 2, Section 4.4), learning rate(Figure 3, Table 3 and Table 14); We will include these interactions between  and other hyperparameters as detail as possible in the paper.
>
> ---
>
> **Q9**: The writing in the paper is confusing. The paper labels standard LoRA as "MAP" without explicit reason.
>
> **A9**: We thank the reviewer for pointing this out. We now been corrected in the revision(Page 1, line 53);  We explicitly explain that “MAP” refers to the deterministic LoRA, and we avoid using this label elsewhere to prevent ambiguity
>
>
> ---
>
> **Q10**: Background on calibration is too brief for non-specialists. Readers unfamiliar with calibration may struggle to interpret the metrics and their practical meaning.
>
> **A10**:  We thank the reviewer for this helpful suggestion. In the revised version, we have expanded the background section on calibration (page 2, line 81)

---

> ### Author Response · Authors · 2025-11-25
> **part 4**
>
> **Q11**: The specific role of each component, the rationale for choosing them, and how they interact are unclear. The method reads like two techniques simply stacked together.
>
> **A11**: Thank you for the comment. Our method is not a simple stacking;it works because **LoRA’s low-rank form matches the SGP’s matrix-normal/Kronecker structure**.
>  because
> LoRA’s low-rank form naturally matches the SGP’s matrix-normal/Kronecker structure.
> LoRA parameterizes the update as
>
> \begin{equation}
> \Delta W = B A .
> \end{equation}
>
> In Sec. 2.2, the sparse-GP model defines the conditional mean of a weight matrix by
>
> \begin{equation}
> M_W(U) = T_r\ U\ T_c, where \ \mathbf{T_r} = Z_r^\top K_r^{-1} \; \mathbf{T_c} = K_c^{-1} Z_c
> \end{equation}
>
> where the projection operators $T_r$ and $T_c$are derived from the inducing
> row/column covariances; Z is covariances matrix.  And it can be treated as $B = T_r\ U$, $A = T_c$ that exactly matches  the LoRa structure.

---

### Author Response · Authors · 2025-11-25
**General Response to All Reviewers**

We thank all four reviewers for their thoughtful feedback. All reviewers pointed out that our experiments were limited in scale; in response, we have added new experiments on **Qwen2.5-14B-Instruct** and **Qwen3-30B-A3B-Instruct-2507**, which we believe substantially strengthen the paper. Their comments greatly improved both the experiments and the overall clarity, and we are very grateful.
* **Reviewer SG6J**: Thank you for emphasizing the importance of calibration and for encouraging a broader empirical evaluation.
* **Reviewer pPT1**: Thank you for appreciating our probabilistic view of LoRA and suggesting evaluations on more modern backbones.
* **Reviewer 3vY8**:We are especially grateful that you carefully checked the technical details and pointed out that the LoRA baseline rank was not clearly specified and the additional comments, all of which were highly valuable in strengthening the paper.
* **Reviewer Sx7d**: Thank you for finding the paper well organized and for urging us to add larger-scale experiments and more detailed comparisons.

---

### Author Response · Authors · 2025-11-28
**Follow-up on Reviews**

Dear Area Chair(s),

First of all, we would like to sincerely thank **you and all reviewers** for your time and efforts on our submission. During the rebuttal phase, we **conducted extensive further experiments** on **Qwen2.5-14B-Instruct** and **Qwen3-30B-A3B-Instruct-2507** and **carefully addressed all reviewer concerns**. One reviewer (pPT1) had already increased their score early in the discussion, and other reviewers might intend to respond to our rebuttal in the final days; however, the system is now locked, and they are no longer able to update their comments and scores.

Across the reviews, the negative scores are associated with **relatively low confidence**, and the positive scores come with **higher confidence**. This makes us believe that, overall, the reviewers do recognize the value of our contribution to the ***Bayesian community***. Our work is already in the process of being merged into the **official PEFT library** as a **LoRA variant**, which we feel further supports its relevance and impact.

Since the reviewers are no longer able to comment, we would be very grateful if **you could briefly look at our paper and rebuttal** and **help ensure that it receives a fair and balanced evaluation**. If you have any concerns of your own, we would be more than happy to address them promptly and thoroughly.

Thank you again for your time.

Best regards,
**All authors**

---

### Meta-Review · Area_Chair_8Mmd · 2026-01-06

**Summary:**

The reviewers raised concerns mainly along three axes. First, on clarity and methodology, several reviewers found the presentation of the Sparse Gaussian Process formulation and its interaction with LoRA and the normalizing flow difficult to follow, initially giving the impression that the method is a stacking of components without sufficiently clear justification. Specific confusion was noted around the role of the inducing variable, how it jointly generates the LoRA factors, and how this differs from existing parameter-space Bayesian LoRA or Laplace-style approaches.

Second, on empirical evaluation, reviewers were concerned that the original experiments were limited in scope (primarily LLaMA-2-7B and a small set of benchmarks) and that the reported improvements in calibration metrics were sometimes modest or inconsistent across tasks. Reviewers requested experiments on larger or more modern models, broader benchmarks, and clearer analysis of the trade-off between predictive quality and inference cost as the number of Monte Carlo samples or flow depth increases.

Third, on presentation details, reviewers pointed out issues such as labeling standard LoRA as “MAP” without explanation and insufficient background for readers unfamiliar with calibration or Gaussian processes.

In the rebuttal and revision, the authors addressed many of these points by adding experiments on larger Qwen-based models, extending evaluations to additional benchmarks, providing quantitative analysis of the effect of the number of MC samples and flow depth, clarifying that a single inducing variable jointly generates the LoRA factors, and improving explanations and terminology. While these changes substantially improve clarity and coverage, some reviewers remained cautious about the overall impact and empirical strength of the gains relative to existing Bayesian PEFT approaches, motivating a conservative recommendation.

**Reviewer Concerns:**

**Concerns largely addressed in the rebuttal**

- **SG6J – W1, W2 (Evaluation scale; MC sampling trade-off):**
  The authors added experiments on larger and more recent models (Qwen2.5-14B-Instruct and Qwen3-30B-A3B-Instruct-2507), including results on the MATH benchmark, addressing concerns about model scale and task diversity. They also provided a quantitative analysis of accuracy, NLL, ECE, and latency as a function of the number of Monte Carlo samples, clarifying the practical accuracy–latency trade-off and justifying the recommended setting of \(S=2\!-\!4\).

- **SG6J – W3 (Flow usefulness):**
  An explicit ablation on flow depth was added, showing that deeper flows yield only modest gains at additional cost, and motivating the default choice of a single-layer flow.

- **SG6J – W3 (Writing: “MAP” terminology; calibration background):**
  The revised version clarifies the use of “MAP” to refer to deterministic LoRA and expands the background on calibration metrics, addressing the main clarity concerns.

- **3vY8 – W2, W4 (Model scale; LoRA rank specification):**
  Additional experiments on larger models address concerns about scalability. The rank of the baseline LoRA model is now explicitly stated and aligned with the Bayesian-LoRA setup.

- **3vY8 – W3, W5 (SGP application to LoRA):**
  The rebuttal clarifies that a single shared inducing variable \(U\) is used to jointly generate the LoRA factors via different projection matrices, and that the SGP is applied to the LoRA update rather than full weight matrices, resolving the main source of confusion between Sec. 2.2, Sec. 3.1, and Algorithm 1.

- **pPT1 – Main concerns:**
  The reviewer explicitly stated that the rebuttal addressed their concerns and increased their score accordingly.

- **Sx7d – Writing clarity; larger models:**
  The explanation of “MAP” terminology and the addition of larger-model experiments address the reviewer’s primary concerns.

---

**Remaining limitations / partially addressed concerns**

- **SG6J – W1 (Magnitude of empirical gains):**
  While the empirical coverage has improved, the observed improvements are generally marginal and not uniformly dominant across all tasks and metrics. This is partly expected given the paper’s focus on calibration rather than optimizing task accuracy, but the practical impact of the gains may still be viewed as limited in some settings.

- **3vY8 – W1 (Method complexity and stacking of components):**
  Although the rebuttal provides clearer explanations and additional analysis, the method still combines multiple components (LoRA, SGP, and a normalizing flow), and their individual contributions are not fully disentangled. Assessing the degree of novelty relative to existing GP-based or Bayesian adaptation methods remains nontrivial.

- **pPT1 – Conceptual concern (function-space vs parameter-space):**
  The rebuttal softened the claims and added discussion, but the advantages of the proposed “function-space” perspective over structured parameter-space Bayesian approaches remain largely conceptual, without targeted experiments or formal analysis demonstrating invariance-related benefits.

**Reviewer Scores:**

- **SG6J:** The reviewer did not provide a post-rebuttal score update. While the rebuttal addressed several concrete concerns (model scale, MC sampling trade-off, and flow ablation), the reviewer did not re-engage in the discussion. A conservative interpretation is that the score remains unchanged at **2 (reject)**.

- **pPT1:** The reviewer explicitly stated in the discussion that their concerns were addressed and that they would increase their score. Based on the original rating (4: marginally below acceptance) and the positive follow-up, a reasonable post-rebuttal score is **6**.

- **3vY8:** This reviewer provided a relatively positive initial assessment (6: marginally above acceptance) with moderate confidence and did not revise their score after the rebuttal. Given that several of their technical clarity concerns were addressed, but no explicit update was made, a conservative assumption is that the score remains at **6**.

- **Sx7d:** The reviewer did not update their score after the rebuttal. Although their main concerns (clarity and larger-model experiments) were addressed, in the absence of an explicit response, a conservative assumption is that the score remains at **4**.

Overall, based on explicit score updates and post-rebuttal comments, a reasonable post-discussion interpretation of the reviews is approximately **6, 6, 4, and 2**, corresponding to a **borderline accept / discuss** profile, with remaining uncertainty driven largely by reviewers with lower confidence who did not participate further in the discussion.

---

### Decision · Program_Chairs · 2026-01-26

Reject